# ROYAL SOCIETY
# OPEN SCIENCE

Subject Areas:
cognition/psychology

Keywords:
voice, face, attractiveness, distinctiveness, averageness, honest signal hypothesis

Author for correspondence:
Romi Zäske
e-mail: romi.zaeske@uni-jena.de

# Attractiveness and distinctiveness between speakers' voices in naturalistic speech and their faces are uncorrelated

Romi Zäske[1,2], Verena Gabriele Skuk[1,2] and Stefan R. Schweinberger[1,3]

[1]Department for General Psychology and Cognitive Neuroscience & DFG Research Unit Person Perception, Institute of Psychology, Friedrich Schiller University of Jena, Am Steiger 3/1, 07743 Jena, Germany
[2]Department of Otorhinolaryngology, Jena University Hospital, Am Klinikum 1, 07747 Jena, Germany
[3]International Max Planck Research School (IMPRS) for the Science of Human History, Max Planck Institute for the Science of Human History, Kahlaische Strasse 10, 07745 Jena, Germany

RZ, 0000-0002-0029-7516

Facial attractiveness has been linked to the averageness (or typicality) of a face and, more tentatively, to a speaker's vocal attractiveness, via the 'honest signal' hypothesis, holding that attractiveness signals good genes. In four experiments, we assessed ratings for attractiveness and two common measures of distinctiveness ('distinctiveness-in-the-crowd', DITC and 'deviation-based distinctiveness', DEV) for faces and voices (simple vowels, or more naturalistic sentences) from 64 young adult speakers (32 female). Consistent and substantial negative correlations between attractiveness and DEV generally supported the averageness account of attractiveness, for both voices and faces. By contrast, and indicating that both measures of distinctiveness reflect different constructs, correlations between attractiveness and DITC were numerically positive for faces (though small and non-significant), and significant for voices in sentence stimuli. Between faces and voices, distinctiveness ratings were uncorrelated. Remarkably, and at variance with the honest signal hypothesis, vocal and facial attractiveness were also uncorrelated in all analyses involving naturalistic, i.e. sentence-based, speech. This result pattern was confirmed using a new set of stimuli and raters (experiment 5). Overall, while our findings strongly support an averageness account of attractiveness for both domains, they provide no evidence for an honest signal account of facial and vocal attractiveness in complex naturalistic speech.

# 1. Introduction

The question of what constitutes attractiveness in a human face or a voice has concerned scientists for a long time (e.g. [1,2]). Contemporary research suggested a substantial role of averageness for perceptions of attractiveness. Specifically, digitally averaged faces are perceived as more attractive the more single faces are entered into an average ([3], but see also [4], on the role of sexual dimorphism). At the same time, averaging decreases distinctiveness by normalizing features that distinguish individual faces from one another. Valentine's influential face space framework [5] holds that all faces we encounter are stored within a multidimensional space relative to an average prototypical face. Accordingly, a given face is more distinctive the more it deviates from the average in terms of its feature dimensions. Similarly, averaging voices across increasing numbers of speakers was more recently reported to enhance perceived vocal attractiveness, at least for simple vocalizations such as vowels or syllables [6,7] and to decrease perceived vocal distinctiveness [8]. In these studies, vocal attractiveness and distinctiveness ratings correlated with the location of voices in a putative multidimensional 'voice-space' in which dimensions reflected acoustic properties such as fundamental frequency (F0; [6–8]), the first formant (F1; [6,7]), formant dispersion [8] or harmonics-to noise ratio [6–8]. Although these effects could in part be owing to a preference for 'typical' stimuli ([9,10], but see [11–13] on the role of sexual dimorphism for vocal attractiveness) and increased typicality by averaging, the relationship between attractiveness and distinctiveness has been rarely studied directly, particularly in the case of voice perception. From an evolutionary perspective, attractiveness has been linked to averageness, symmetry and sexual dimorphism. Facial averageness and symmetry, in particular, may be perceived as attractive, as they signal reproductive viability, developmental stability and health [14]. Similarly, vocal attractiveness has been linked to averageness in morphed voices [7] and in natural voices, in terms of a negative relationship with acoustic distance-to-mean, i.e. deviation from averageness (cf. supplemental Fig. S3 of [15]). From an evolutionary perspective, average voices may be perceived as more attractive than non-typical voices, in the sense that deviation from the norm could signal ill-health (e.g. [16]), which is thought to be negatively related to physical attractiveness [17]. Furthermore, from a perceptual point of view, several researchers argued that average (typical) stimuli, including voices [9], are processed more fluently and are therefore perceived as more beautiful or likeable than non-typical ones [18]. Thus, while average voices are also attractive, a role of symmetry for the voice domain is less obvious. Nevertheless, it is feasible to assume that asymmetrical shape or size of the vocal folds can cause irregular vibration and dysphonia, i.e. hoarse voice quality [19], as a possible indicator of various diseases (for review, see [20,21]) and hormonal status in women [22,23].

Of relevance, the relationship of vocal attractiveness and distinctiveness appears to be affected by the speech material used. Specifically, while for simple vowel stimuli perceived attractiveness was reported to correlate negatively with the deviation of acoustic measures from the average [6,7], no correlation between these measures and attractiveness ratings was found for more complex word or sentence stimuli [6]. On the one hand, this may be surprising considering that other speaker attributes can usually be discerned across various utterances of a given speaker, such as identity [24], sex [25,26] or age [27]. On the other hand, the processing of linguistic and non-linguistic properties of voices, i.e. speech content and speaker attributes, may not be entirely independent (for recent reviews, see [28,29]). In the present study, we assessed relationships of perceived attractiveness and distinctiveness in faces and voices while using two different types of speech materials (vowels versus more complex naturalistic sentences).

In the case of faces, distinctiveness has been assessed by two measures: One measure of distinctiveness (face-in-the-crowd, or FITC distinctiveness) involves asking raters how easily they would spot a face in a group of people (e.g. [30–32]). A second measure (deviation-based, or DEV distinctiveness) involves asking raters to judge how much a face deviates from other known faces (e.g. [33]). Although these measures are often taken to reflect the same construct, a recent study on a large set of 560 faces, rated for both types of distinctiveness, suggested that both measures only correlate to a very moderate extent [34]. Strikingly, while there was a substantial negative correlation between DEV distinctiveness and attractiveness, there was a small positive correlation between FITC distinctiveness and attractiveness. A similarly detailed study on the relationship between distinctiveness and attractiveness of voices is unavailable as yet. One interpretation of the above findings suggests that the deviation-based measure of distinctiveness is the more valid measure in terms of Valentine's model [5], and that the FITC rating might be distorted by cognitive heuristics such as 'Surely I would spot such an attractive face', thus biasing FITC ratings for extremely attractive faces towards higher FITC distinctiveness [34]. If so, we would expect analogous patterns of

relationships between corresponding measures of vocal distinctiveness and attractiveness (i.e. substantial negative correlations of vocal attractiveness with DEV distinctiveness, and no such pattern, or even small positive correlations with voice-in-the-crowd (VITC) distinctiveness).

Perceived attractiveness is often taken as a signal for genetic fitness or health across sensory modalities (e.g. [35,36–38]). For instance, physical features of both faces and voices could be influenced by common (e.g. hormonal) factors, and accordingly might provide 'honest signals' to mate value [35,39]. If so, such common factors could provide a basis for positive correlations between facial and vocal attractiveness in the same individual. Indeed, a few studies found vocal and facial attractiveness of women to be correlated [40–42]. Moreover, vocal and visual attractiveness of 12 male speakers were also found to be correlated when rated by either adolescents or adults (but not when rated by children: [33]), and for 60 male speakers rated by female listeners [44]. Others found a correlation only after aggregating data across female and male speakers [45]. In the individual studies, voice ratings were either based on vowels or single words [41,43,45] or on a single sentence [46] or a short text passage [42,44], such that the relationship between vocal and facial attractiveness has never been compared for different types of utterances. Note that two studies suggest that positive correlations between ratings for faces and voices of women [47] and men [48] could be based on sexual dimorphism, rather than on prototypicality or averageness *per se*, as an aspect of attractiveness (cf. [14]). The consideration of utterance type in voices may be important because isolated vowels and sentences differ substantially in the amount and type of phonetic information they convey. Sentences in particular convey information not only about biophysical inevitabilities of an individual speaker but also about socially acquired voice characteristics which relate to, for instance, phonetic or prosodic differences between male and female voices [49]. Therefore, it is reasonable that the type of utterance also affects personal impressions of attractiveness and distinctiveness (for a similar argument, see [9]).

In the present study, we obtained photographs and voices from a set of young adult speakers, and recorded ratings of attractiveness and two measures of distinctiveness both for vowel stimuli (experiment 1 and 2) and sentence stimuli (experiment 3 and 4). Our first aim was to determine whether a pattern of relationships between measures of distinctiveness and attractiveness that is similar to what has been reported recently for faces [34] would also emerge for voices. Our second aim was to investigate the degree of cross-domain relationships between measures of attractiveness and distinctiveness for the voices and faces of the same speakers, while considering potential effects for different types of utterances. Note that we assessed ratings of faces and voices when these were presented in isolation, rather than in a multimodal situation, to avoid the possibility of inflated degrees of similarity of ratings for corresponding faces and voices that could be caused by multisensory integration [50].

## 2. Experiments 1–4

### 2.1. Methods

#### 2.1.1. Recording

*Speakers.* We recruited 64 young speakers (32 female, $M = 21.9$ years, aged 18–25 years) among students of the University of Jena. Age of female and male speakers did not differ significantly ($t_{62} = -0.392$, $p = 0.696$). All speakers received €10 for participation and gave their written informed consent to the use of their faces and voices as stimuli in the current and future experiments. Based on previous reports of positive correlations between facial and vocal attractiveness in the range of $r = 0.37$–0.60 for $n = 30$–42 voices and faces [40,41,46], we conducted a power analysis using G-Power 3.1 [51] to determine the minimum sample size required for detecting a medium-sized effect of 0.40 (one-tailed) with a power of 0.80 and an $\alpha$ error probability of 0.05. This analysis revealed a minimum sample size of $n = 34$. Accordingly, we deemed a sample size of 64 voices and faces overall appropriate to detect medium-sized ($r \sim 0.40$) cross-modal correlations with adequate power, even when analysing data for female and male speakers ($n = 32$ each) separately.

*Stimuli.* From each speaker, we took photographs and recorded sustained vowels ([a:],[e:],[i:],[o:],[u:]) and German sentences (sentence no. 1: 'Keine Antwort ist auch eine Antwort' [no answer is an answer too.]; sentence no. 2: 'Magdala liegt bei Apolda' [Magdala is near Apolda.]; sentence no. 3: 'Die Oma mag Urlaub am Meer.' [grandma likes holidays by the sea.]) in a quiet and semi-anechoic room, as part of a more extensive protocol. The recording room was situated in a quiet tract of the building and did not

have any windows. Walls, ceiling and floor of this room were fitted with molleton, to eliminate reverberation. To standardize photographs, speakers sat on a chair in front of a green background and were illuminated by a three-point lighting system. All wore a black cape around their shoulders, took off glasses, jewellry and make-up and men shaved before the session. They were further instructed to face the camera and to show an emotionally neutral expression to minimize potential effects of changeable face characteristics on inter-speaker variability, in analogy to voice recording procedures (see below). Portraits were captured with a Sony DCR-DVD403E camcorder. Using Adobe Photoshop CS5™, faces were cropped, standardized in size and vertical eye position, and pasted on a grey background.

To reduce influences of momentary voice states' (such as a specific emotional expression or speaking style), and to ensure that between-speaker variation was instead characteristic for more robust individual voice traits', speakers were presented with a pre-recorded speaker (first author, presented via loudspeakers), and asked to produce vowels and sentences naturally but in emotionally neutral intonation, and in similar style, intensity and timing. Voices were recorded with a Sennheiser MD 421-II microphone with a pop protection and a Zoom H4n audio interface (16-bit resolution, 48 kHz). The interface was connected to a computer in the neighbouring room in which the audio manager monitored recordings via Adobe Audition 3.0. Each utterance was recorded four to five times to choose the best audio recordings (no artefacts and clear articulation) as stimuli. Using PRAAT [52], vowels were cut to contain the stable phase (duration 1500 ms) and sentences were cut at first-word onset and last word offset (sentence durations: no. 1: $M = 2358$ ms, s.d. = 245, range 1833–3081 ms; no. 2: $M = 1979$ ms, s.d. = 180, range 1583–2671 ms; no. 3: $M = 1883$ ms, s.d. = 133, range 1620–2207 ms). All stimuli were converted to mono, and were root mean square (RMS) normalized to 70 dB. In order to fade vowel stimuli in and out the first and last 180 ms were multiplied with a cosine window function.

### 2.1.2. Rating

*Raters*. Four independent and non-overlapping groups of twenty (10 female) student participants each contributed data to experiments 1–4, respectively (experiment 1: four left-handed, $M = 21.7$ years, range 19–28; experiment 2: all right-handed, $M = 23.5$ years, range 18–30; experiment 3: six left-handed, $M = 22.4$ years, range 18–30; experiment 4: two left-handed, $M = 22.3$ years, range 19–27). Data from five additional participants were excluded from the analyses (experiment 1: $n = 3$; experiment 3: $n = 1$; experiment 4: $n = 1$) due to technical problems or task incompliance ($n = 3$), or extremely low variance of responses (s.d. < 0.60; $n = 2$). All participants were native speakers of German and none reported hearing difficulties. Participants received a payment of €5 or course credit. All gave written informed consent. The study was conducted in accordance with the Declaration of Helsinki, and was approved by the Faculty Ethics Committee of the University of Jena.

*Procedure*. Raters were tested individually in a quiet sound-attenuated chamber, situated within the experimenter's room. Instructions were presented in writing on a computer screen. Voice stimuli were presented diotically via Sennheiser HD 212Pro over-ear headphones at a constant and comfortable intensity, with an approximate peak intensity of 70 dB(A) as determined with a Brüel & Kjær Precision Sound Level Meter Type 2206. Face stimuli subtended a visual angle of approximately 10.9° vertical × 7.9° horizontal, at a viewing distance of 65 cm. Voices and faces were rated blockwise on 6-point rating scales, with sub-blocks for attractiveness ratings and distinctiveness ratings. Depending on the experiment, distinctiveness was measured as either 'distinctiveness-in-the-crowd' (DITC) for voices/faces (VITC/FITC, experiments 1 and 3), or as deviation (DEV) from the mean (experiments 2 and 4). Individual trials used single presentations either of one of five *vowels* (experiments 1 and 2), or of one of three *sentences* (experiments 3 and 4) per speaker. Sub-blocks for attractiveness and distinctiveness ratings were further subdivided into male and female stimulus blocks, each of which contained either 32 face trials (i.e. one trial per speaker) or a number of voice trials depending on the experiment: while there were 160 voice trials (32 speakers × 5 vowels) per male and female rating block in experiments 1 and 2, 96 trials were presented in experiments 3 and 4 (32 speakers × 3 sentences). Within male and female blocks, trials were always presented in randomized order. Across participants, block order was balanced for modality (face or voice), rating task (attractiveness or distinctiveness) and stimulus sex (male or female). Specifically, one group of participants first rated faces for attractiveness starting with male faces, followed by female faces, then rated distinctiveness of male and female faces, and finally performed voice ratings in the analogous block order as for faces. In another group, the block order was swapped, such that participants first rated voices for distinctiveness starting with female voices, followed by male voices, then rated attractiveness of female and male voices, and finally rated faces according to the same block order. Rating instructions

for the various tasks were as follows (with German translations in parentheses): attractiveness: 'please assess how attractive/unattractive the voices are, in the sense of sounding pleasant' (Beurteilen Sie bitte nun, wie attraktiv/unattraktiv die Stimmen sind. Attraktivität ist hier gemeint im Sinne von `wohlklingend'.) or 'please assess how attractive/unattractive the faces are, in the sense of good-looking' (Beurteilen Sie bitte nun, wie attraktiv/unattraktiv die Gesichter sind. Attraktivität ist hier gemeint im Sinne von 'gutaussehend'.); VITC: 'please assess how distinctive the voices are. Imagine yourself on a busy square. You are surrounded by many people who are talking simultaneously. A voice is distinctive if it stands out of the crowd' (Beurteilen Sie bitte die Stimmen danach, wie distinkt sie klingen. Stellen Sie sich dazu vor, dass Sie auf einem belebten Platz stehen. Sie sind umgeben von vielen Personen, die durcheinander reden. Eine Stimme ist distinkt, wenn sie aus der Menge heraussticht.) or FITC: 'please assess how distinctive the faces are. Imagine yourself on a busy square. You are surrounded by many people. A face is distinctive if it stands out of the crowd' (Beurteilen Sie bitte die Gesichter danach, wie distinkt sie aussehen. Stellen Sie sich dazu vor, dass sie auf einem belebten Platz stehen. Sie sind umgeben von vielen Gesichtern. Ein Gesicht ist distinkt, wenn es aus der Menge heraussticht.). DEV: 'please assess how typical/atypical the voices/faces are. Ask yourself how much the respective voice/ face differs from other voices/faces you know.' (Beurteilen Sie bitte die Stimmen/Gesichter danach, wie typisch/untypisch sie sind. Fragen Sie sich dazu, wie stark sich die jeweilige Stimme/das jeweilige Gesicht von anderen Ihnen bekannten Stimmen/Gesichtern unterscheidet.)

With respect to the exact wording of instructions, it should be noted, that while in most previous studies participants were simply asked to rate attractiveness of a given stimulus, without further specification (i.e. attractiveness in the sense of sexual attraction, likeability, aesthetics, or beauty), we decided to disambiguate the term for our participants. By adding the synonyms 'good-looking' and 'sounding pleasant', we reasoned that participants would refer to a similar concept that would instantly make sense for the respective modality: e.g. whereas a person can be attractive, his or her face may be 'good-looking', and his or her voice may 'sound pleasant'. Note that, although the precise definition of concepts related to attractiveness (including beauty, or pleasantness) remains a notorious issue in the field, attractiveness and pleasantness are correlated (e.g. [53]). In terms of distinctiveness, instructions were based on the face literature [30–33] and adapted for the voice domain.

The experiment lasted 60 min. Trials started with a black central fixation cross on a grey background with a reminder of the task and response alternatives below. After 1000 ms, the cross was either joined by a voice with a constant duration for vowels in experiments 1 and 2 (1500 ms) and variable duration for sentences in experiments 3 and 4 (cf. §2.1.1 Stimuli), or replaced by a face which remained on screen until a response was entered. This means that whereas voice stimuli were presented for the duration of the utterance (between about 1.5 and 3.1 s, depending on utterance), face stimuli remained on screen until a response was given. Note that after about 100 ms of exposure duration, evaluations of faces appear to remain remarkably stable even for much longer exposure durations [54]. Accordingly, while inherent differences in timing at which social information unfolds in static faces when compared with voices preclude a fair comparison, such procedural differences are unlikely to have affected the present findings. There was no time limit for the responses, but participants were encouraged to respond spontaneously and as accurately as possible. Ratings (six-point scales, with 1 = very unattractive/undistinctive/typical and 6 = very attractive/distinctive/atypical). Finally, we recorded familiarity judgements for faces, to rule out prior familiarity with the speakers. Trial procedures were analogous to the preceding blocks. Responses in this 2-alternative forced choice familiarity judgement task ('d' or 'l' for familiar versus unfamiliar) were entered via a computer keyboard.

## 2.2. Results and discussion

For each experiment, we computed Spearman rank correlations between mean attractiveness and distinctiveness ratings per speaker (averaged across raters) within voice and face domains (cf. figure 1), as well as between domains (cf. figure 2). Non-parametric rank correlations were computed because these should be robust to differences in how the rating scales are used in different experimental conditions, rating dimensions, or participants. Based on face familiarity judgements, we excluded a small percentage of ratings for familiar speakers from these analyses (3.4%, 2.0%, 1.5% and 1.1% of trials for experiments 1–4, respectively). On average, each stimulus thereby obtained 19.3/20 (s.d. = 0.9, range 15–20), 19.6/20 (s.d. = 0.6, range 18–20), 19.7/20 (s.d. = 0.5, range 18–20) and 19.8/20 (s.d. = 0.5, range 18–20) of valid ratings in experiments 1–4, respectively. For the calculation of interrater-reliability, i.e. the agreement between raters on attractiveness and distinctiveness of faces and voices, Cronbach's $\alpha$ was computed (with ratings for each speaker collapsed across individual vowels or sentences). To this

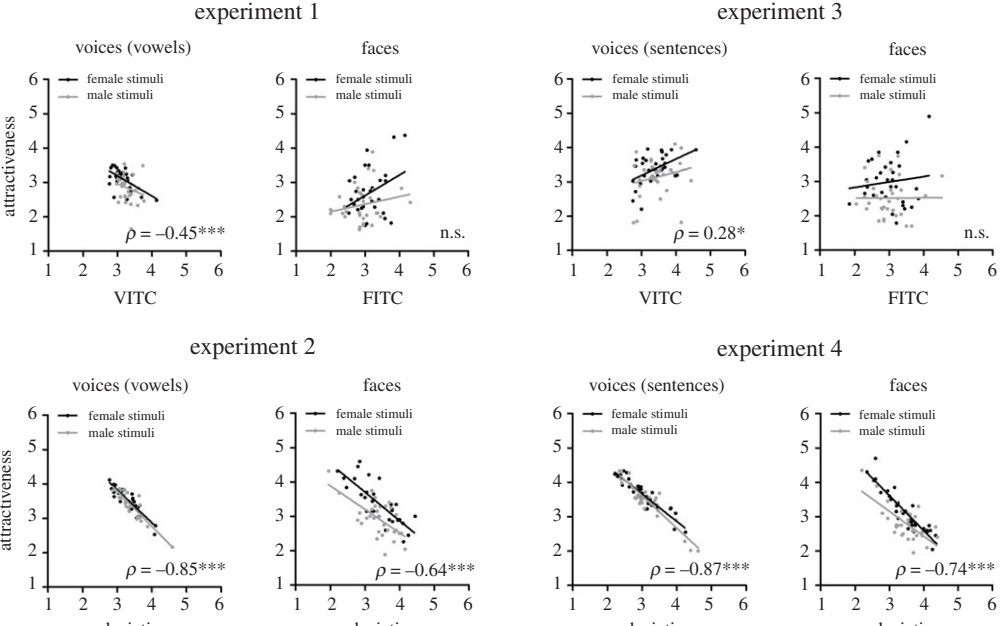

**Figure 1.** Within-domain correlations between mean ratings of attractiveness and distinctiveness (top: DITC-based distinctiveness; bottom: deviation-based distinctiveness) for faces and voices based on vowels (left: experiments 1 and 2) and based on sentences (right: experiments 3 and 4). Data points represent individual speakers, and are depicted separately for female (black) and male speakers (grey). Correlation coefficients ($\rho$) are depicted overall, with asterisks indicating significance levels (* $p < 0.05$; ** $p < 0.01$; *** $p < 0.001$, uncorrected). Note that the strong and negative correlations between attractiveness and DEV-based distinctiveness (experiments 2 and 4) survived Bonferroni correction for the 48 tests depicted in figures 1 and 2 (electronic supplementary material, tables S1 and S2), with the exception of the correlation for male faces in experiment 2 (corrected alpha level = 0.001). The only other correlation which failed to reach significance after Bonferroni correction was the small positive correlation between voice attractiveness and VITC across all speakers (experiment 3).

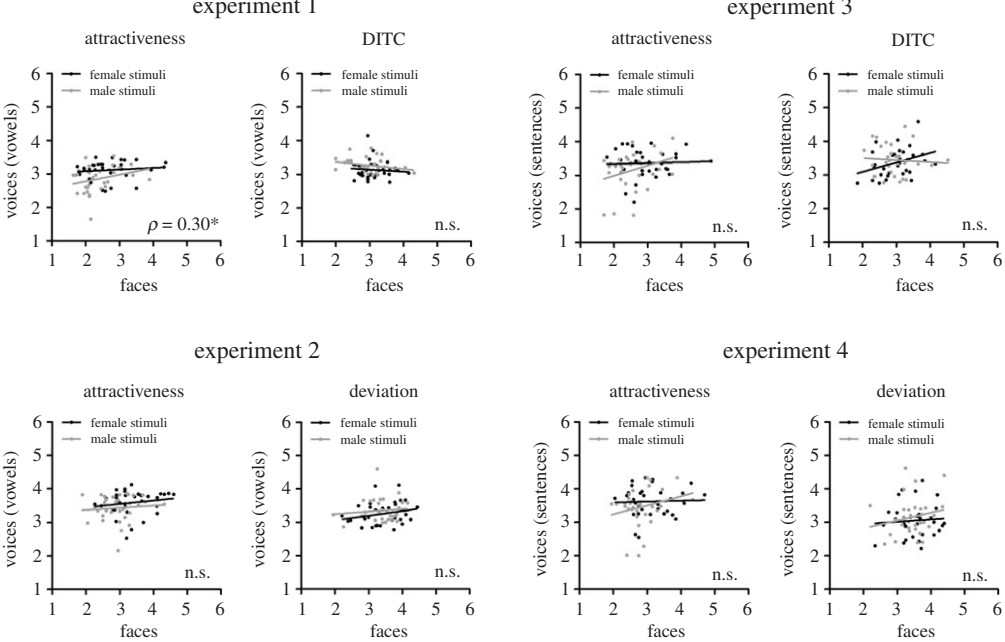

**Figure 2.** Between-domain correlations between mean ratings of faces and voices based on vowels (left: experiments 1 and 2) and based on sentences (right: experiments 3 and 4). Data are depicted separately for attractiveness and distinctiveness ratings (top: DITC-based distinctiveness; bottom: deviation-based distinctiveness) and for female (black) and male speakers (grey). Correlation coefficients ($\rho$) are depicted overall, with asterisks indicating significance levels (* $p < 0.05$; ** $p < 0.01$; *** $p < 0.001$, uncorrected). Note that facial and vocal attractiveness ratings for the same speakers were generally uncorrelated, with the exception of a small positive correlation when voice attractiveness ratings were based on vowel stimuli (experiment 1). This correlation was no longer significant after Bonferroni correction for the 48 tests depicted in figures 1 and 2 (corrected alpha level = 0.001).

end, ratings for speakers with whom raters had indicated familiarity (3.4%, 2.0%, 1.5% and 1.1% of trials for experiments 1–4, respectively) were replaced by mean ratings per participant across all speakers (thus, leading to more conservative, i.e. lower reliability estimates). Interrater-reliability was generally high ($0.763 \leq \alpha < 0.923$) except for DITC ratings for voices based on vowel utterances ($0.640 \leq \alpha < 0.647$, cf. electronic supplementary material, table S3).

For correlations within and between voice and face domains, voice ratings were averaged across the five vowels (experiments 1 and 2) and across the three sentences (experiments 3 and 4). Owing to technical issues one sound sample (sentence no. 1) of a female speaker (fKL23) in experiments 3 and 4 failed to be presented, such that mean ratings of that speaker's voice were based on two, instead of three, sentences. This was done to aggregate across more trials in order to obtain more stable ratings per speaker, and because initial analyses had suggested consistently positive correlations between both individual vowels, and between individual sentences.

Within domains (electronic supplementary material, table S1; figure 1), we obtained a significant negative correlation between DITC and attractiveness ratings in experiment 1 when ratings for voices were based on vowels ($\rho_{62} = -0.45$, $p < 0.001$). Accordingly, voices rated as more distinctive in VITC ratings were rated as less attractive. For faces, no negative correlation between FITC and attractiveness was found; to the contrary, we observed a marginal positive correlation ($\rho_{62} = 0.21$, $p = 0.101$). In experiment 2, strong negative correlations between DEV and attractiveness were seen both for voices based on vowels ($\rho_{62} = -0.85$, $p < 0.001$) and for faces ($\rho_{62} = -0.64$, $p < 0.001$). In experiment 3, a small positive correlation between DITC and attractiveness was seen for voices based on sentences ($\rho_{62} = 0.28$, $p = 0.028$), but again no significant correlation emerged for faces ($\rho_{62} = -0.06$, $p = 0.660$). In experiment 4, strong negative correlations between DEV and attractiveness were seen both for voices based on sentences ($\rho_{62} = -0.87$, $p < 0.001$) and, as in experiment 2, for faces ($\rho_{62} = -0.74$, $p < 0.001$).

To explore possible effects of the type of utterance on the above within-domain correlations, we also computed correlations for voices separately for each vowel (experiments 1 and 2) and sentence (experiments 3 and 4). Details can be found in the electronic supplementary material, tables S4–S7. In short, these analyses indicate that the negative correlations between voice attractiveness and DEV were consistently present across vowels and sentences throughout (electronic supplementary material, tables S5 and S7, experiments 2 and 4). By contrast, both the negative correlations between voice attractiveness and DITC for vowels (electronic supplementary material, table S4, experiment 1) and the positive correlations between voice attractiveness and DITC for sentences (electronic supplementary material, table S6, experiment 3) were less consistently seen across voice samples, suggesting a role of individual utterances in addition to an effect of speaker *per se*. For the interested reader, we report within-domain correlations also separately for male and female raters (electronic supplementary material, tables S12 and S13). In short, these analyses suggest similar results for both male and female raters. We also report statistical comparisons of the magnitudes of correlations for female versus male speakers (electronic supplementary material, table S16), which generally show similar correlations as well. The only exception was significantly larger negative correlations between face attractiveness and DEV for female compared to male faces, in both experiments 2 and 4. Of note, negative correlations remained significant and highly consistent for male stimuli when tested separately, indicating a qualitatively similar, albeit relatively weaker, relationship.[1]

Between domains (electronic supplementary material table S2; figure 2), we generally observed no significant correlations between vocal and facial attractiveness for experiments 2–4 ($0.09 < \rho_{62} < 0.24$; $0.059 < p < 0.476$). The only exception was a small but significant positive correlation between vocal and facial attractiveness ratings in experiment 1, when voice ratings were based on vowel stimuli ($\rho_{62} = 0.30$, $p = 0.014$).[2] Moreover, no significant between-domain correlations were found for distinctiveness ratings in any of the experiments ($-0.21 < \rho_{62} < 0.18$; $0.095 < p < 0.476$). Between-domain correlations were also computed separately for individual vowels and sentences in the respective experiments. Electronic supplementary material, tables S8–S11 suggests that the absence of correlations between vocal and facial attractiveness and distinctiveness described above held across individual voice stimuli.

When the above analyses were conducted separately for male and female speakers, we obtained virtually the same pattern of results (cf. electronic supplementary material, tables S1 and S2) with a

---

[1]Note that the strong and negative correlations between attractiveness and DEV-based distinctiveness (experiments 2 and 4) survived Bonferroni correction for the 48 tests depicted in figures 1 and 2 (electronic supplementary material, tables S1 and S2), with the exception of the correlation for male faces in experiment 2. The only other correlation which failed to reach significance after Bonferroni correction was the small positive correlation between voice attractiveness and VITC across all speakers (experiment 3).

[2]This correlation was no longer significant, however, after Bonferroni correction for the overall 48 tests depicted in figures 1 and 2 (electronic supplementary material, tables S1 and S2).

few exceptions: in experiment 1 the within-domain correlation between voice attractiveness and DITC was reduced to insignificance for male voices ($\rho_{30} = -0.32$, $p = 0.070$), as were between-domain correlations for attractiveness when female and male stimuli were analysed separately. In experiment 3, the within-domain correlation between voice attractiveness and DITC was absent for male voices ($\rho_{30} = 0.12$, $p = 0.515$). However, it should be noted that these deviations from the overall pattern of results probably reflect a reduction of statistical power owing to limited sample size. For the interested reader, we report between-domain correlations also separately for male and female raters (electronic supplementary material, tables S14 and S15). In short, these analyses suggested similar results for both male and female raters. We also report statistical comparisons of the magnitudes of correlations for female versus male speakers (electronic supplementary material, tables S16 and S17). In short, these analyses did not suggest differences either.

Please note that although we decided not to correct for multiple correlations to avoid false-negative conclusions, our main finding of strong and negative correlations between attractiveness and DEV-based distinctiveness survived Bonferroni correction for overall 48 tests in experiments 1–4, with a corrected alpha level of 0.001, with the exception of one correlation for male faces (as specified in the electronic supplementary material, table S1). We considered that a problem of using an alpha adjustment for multiple tests (such as the Bonferroni correction) could have been that we probably would have missed small but important effects reported by others, such as small positive correlations between voice and face attractiveness in possible support of the honest-signal hypothesis [41,43]. In fact, although the small positive correlations between facial and vocal attractiveness for vowel stimuli correspond to a specific hypothesis and previous published research, note that these correlations would not have survived a Bonferroni correction (cf. electronic supplementary material, table S2).

One possible limitation of these four experiments could be our use of a model speaker during voice recordings. Although we did this to reduce variability of undesired origin (such as variability in emotional expression or speaking style), we cannot fully exclude the possibility that instructions to pronounce the utterances in similar style, intensity and timing as the model could have constrained speakers in expressing their natural vocal variability, which in turn could have compromised our ability to find correlations between voice and face attractiveness.[3] As another possible limitation, our instructions for attractiveness ratings differed slightly between faces and voices, in that we further specified attractive as 'good-looking' and 'sounding pleasant', respectively. To address both these issues, we conducted a fifth experiment for which we recorded an entirely new set of 21 speakers producing two sentences and vowels, both without and with a model speaker (experiment 5). An independent group of listeners then rated the speakers' faces and voices for attractiveness and DEV. In line with our reasoning that our recording procedure from experiments 1–4 should preserve idiosyncratic and stable vocal traits, essentially we hypothesized that we would replicate our results from experiments 2 and 4 *independent* of recording mode (with versus without a model speaker). First, we predicted that the correlation between vocal attractiveness and DEV would be substantial and negative for both utterance types (sentences and vowels) and independent of recording modes (with and without model speaker). Second, if the model speaker preserves natural voice quality, vocal attractiveness ratings should correlate positively between different sentences (S1 and S2), and these correlations should not be smaller between recording modes versus within recording modes. Third, we predicted to find no evidence for a correlation of vocal and facial attractiveness ratings for sentences, neither for sentences recorded with nor without the model. Finally, we predicted that the acoustic variability (in terms of vowel pitch and sentence duration) would be higher for samples recorded without, compared to with, the model speaker. Note, that we pre-registered these hypotheses along with the study design on the Open Science Framework (OSF) platform before we collected data [55].

# 3. Experiment 5

## 3.1. Methods

### 3.1.1. Speakers

Face and voice stimuli were recorded of 21 novel native speakers of German (12 female), aged 18–28 years ($M = 21.7$) as recruited among students of the University of Jena. A prior sensitivity analysis

---

[3]We thank an anonymous reviewer for directing our attention to this point.

using G*Power V.3.1.9.7 suggested that with a sample size of $n = 21$, an $\alpha$ error probability of 0.05 (one-tailed), and a power of 0.80, the test would be sensitive to correlations from $|\rho| = 0.49$. This was considered sufficient to replicate the main findings of experiments 2 and 4, with significant negative correlations ($-0.87 \leq \rho \leq -0.64$) between attractiveness and DEV for speech produced after a model speaker. For the same analyses on speech produced without a model, we predicted similarly sized effects.

Recording procedures and stimulus editing of audio files and images were analogous to experiments 1–4, except that we recorded two versions of the same four utterances, as part of a more extensive and standardized protocol. The utterances were two of the German sentences (Die Oma mag Urlaub am Meer.' and 'Keine Antwort ist auch eine Antwort.) and vowels (/a/ and /i/), also recorded for experiments 1–4. In the first part of the recording session, the voice stimuli were recorded following instructions to produce a written sentence or sustained vowel three times (without model speaker). Importantly, written speech material was never verbally mentioned to speakers, to allow for natural variance between speakers. Speakers were first shown the respective sentence or vowel, then looked straight into the camera and produced each utterance from memory. Note that we chose this procedure to approximate 'free' speech despite standardized content, and yet largely avoid read speech. In the second part of the recording session, we re-recorded the same utterances using the model speaker and the same instructions as in experiments 1–4.

### 3.1.2. Raters

Twenty (10 female) student participants, who had not participated in experiments 1–4, contributed data to experiment 5 (1 left-handed, $M = 21.4$ years, range 18–26 years). All were native speakers of German and did not report hearing difficulties. Ratings considered for analyses were based on unfamiliar speakers, as controlled via a post-experimental familiarity task with static faces of the speakers. Accordingly, data of one further participant who indicated familiarity with 17 of the 21 speakers had to be excluded. Of the final sample, 17 raters were unfamiliar with all speakers, and three raters were familiar with one, two, and three speakers, respectively. The respective data points were replaced with the mean ratings of a given rater for each condition. Raters received course credit and chocolate for participation. The study was conducted in accordance with the Declaration of Helsinki, and has been approved by the Faculty Ethics Committee of the University of Jena.

### 3.1.3. Procedure

All experimental procedures were analogous to experiments 1–4, with the following exceptions. Rating instructions for attractiveness were identical for face and voice trials (please assess how attractive/unattractive the voices/faces are). Sentences and vowels were both rated by the same group, and thus became a within-subjects factor (utterance type). Overall, there were 12 sub-blocks (four face blocks and eight-voice blocks) for the rating tasks across which speaker identities were repeated. Block order was counterbalanced in two experiment versions and organized in the same blocking hierarchy for the first (and second) experiment version: participants first rated faces (or voices), attractiveness (or DEV), male (or female) speakers and vowel (or sentence) stimuli (for voice blocks only). Block order was counterbalanced across male and female raters. Within the sub-blocks all stimuli were presented in random order. For the four face sub-blocks (2 rating dimensions × 2 speaker genders), each face was presented exactly once per rating dimension, resulting in 42 trials overall (21 speakers × 2 rating dimensions). In each of the eight-voice sub-blocks (2 rating dimensions × 2 speaker genders × 2 utterance types), each speaker was presented four times. This number results from the two recording modes (with and without model speaker) and two different utterances per vowel block (/a/ and /i/) and per sentence block (S1: 'Die Oma mag Urlaub am Meer.' And S2: 'Keine Antwort ist auch eine Antwort.'). Like in experiments 1–4, the final task was to categorize the faces of all 21 speakers according to pre-experimental familiarity. Including individual breaks after every approximately 4 min, the experiment lasted approximately 30 min.

### 3.2. Results

Data preparation was analogous to experiments 1–4, with the only difference, that ratings were collapsed across two sentences (instead of three) and two vowels (instead of five), respectively. We did not correct for multiple tests because tests refer to pre-registered hypotheses, unless they are labelled as exploratory.

To address our first prediction of negative correlations between vocal attractiveness and DEV, we computed Spearman rank correlations (one-tailed) between both measures, separately for the two recording modes (with and without model) and two utterance types (vowels and sentences). All correlations were negative, for both vowels (with model: $\rho_{19} = -0.82$, $p < 0.001$; without model: $\rho_{19} = -0.46$, $p = 0.018$), and sentences (with model: $\rho_{19} = -0.89$, $p < 0.001$; without model: $\rho_{19} = -0.59$, $p = 0.003$). Comparisons of correlations (two-tailed) between recording modes were performed using a statistical package for R [56]. Results based on Steiger's $z$ [57], indicated larger correlations between vocal attractiveness and DEV for voices recorded with a model speaker when compared with voices recorded without the model (vowels: $z = -2.115$, $p = 0.035$; sentences: $z = -2.974$, $p = 0.003$). To further explore this unpredicted finding, we also calculated correlations across recording modes, separately for each utterance type and rating dimension. Ratings across recording modes correlated positively and significantly for both, attractiveness (vowels: $\rho_{19} = 0.72$, $p < 0.001$; sentences: $\rho_{19} = 0.79$, $p < 0.001$) and DEV (vowels: $\rho_{19} = 0.38$, $p = 0.046$; sentences: $\rho_{19} = 0.75$, $p < 0.001$). There was a trend for a smaller correlation between recording modes for DEV compared to attractiveness ratings for vowels ($z = 1.760$, $p = 0.079$), but not for sentences ($z = 0.317$, $p = 0.751$).

Regarding the second hypothesis of positive attractiveness correlations between the two sentences independent of recording mode, we first computed Spearman rank correlations (one-tailed) between attractiveness ratings for sentences S1 and S2, both within and across recording modes. As predicted, all between-sentence correlations were positive and significant, both within recording modes (with model: $\rho_{19} = 0.77$, $p < 0.001$; without model: $\rho_{19} = 0.80$, $p < 0.001$) as well as across recording modes (S1[with model] × S2[without model]: $\rho_{19} = 0.60$, $p = 0.002$; S1[without model] × S2[with model]: $\rho_{19} = 0.64$, $p = 0.001$). Comparisons of the two within-modes sentence correlations with each of the two across-modes sentence correlations did not indicate significant differences (all $1.228 \leq z \leq 1.769$; all $0.077 \leq ps \leq 0.220$).[4]

In line with our third hypothesis, we found no evidence for correlations (two-tailed) of facial and vocal attractiveness, neither for sentences recorded with the model ($\rho_{19} = 0.25$, $p = 0.279$), nor without the model ($\rho_{19} = 0.06$, $p = 0.808$). An exploratory analysis of the same correlation for vowels, yielded no significant results either (with model: $\rho_{19} = -0.30$, $p = 0.179$; without model: $\rho_{19} = -0.18$, $p = 0.427$).

Finally, regarding the hypothesis that acoustic variability will be higher for samples recorded without a model speaker compared to samples with a model speaker, we first determined F0 in both vowels (/a/ and /i/) and duration of both sentences (S1 and S2) by means of PRAAT [52], cf. electronic supplementary material, tables S18 and S19 for acoustic measurements. To this end, we measured mean pitch across the entire duration of the respective vowel (1500 ms), using an automated script with 50 Hz and 500 Hz as lower and upper boundaries and with the time steps setting kept at default (0.0). Note, that vowel stimuli had been cut to contain the stable phase of the original vowel recordings (cf. methods). Data were then collapsed across vowels and across sentences, respectively, to obtain one data point per utterance type, recording mode and speaker. To statistically compare the variances between recording modes, we used a common method for dependent samples [58]. Specifically, we first calculated the sum and the difference between the ratings of the two recording modes, then correlated these two measures to test if this correlation differs from zero, using a student $t$-test. A significant result would indicate that variances differ between the two recording modes. The same test was performed on vowel F0 and sentence duration data. The Pearson correlation between the sum and the difference of vowel F0 in the two recording modes ($r_{19} = 0.15$, $p = 0.519$) did not significantly differ from zero, $t_{19} = 0.621$, $p = 0.271$ (one-tailed). The same held for the correlation for sentence duration ($r_{19} = -0.03$, $p = 0.915$), $t_{19} = -0.103$, $p = 0.46$ (one-tailed). This suggests that variances in vowel pitch and sentence duration do not differ between recording modes (also cf. electronic supplementary material, tables S18 and S19).[5]

[4]Following up on a reviewer request, we performed analogous analyses on DEV ratings, as exploratory analyses. We obtained the same result pattern as for attractiveness, i.e. between-sentence correlations were positive and significant, both within recording modes (with model: $\rho_{19} = 0.65$, $p = 0.001$; without model: $\rho_{19} = 0.79$, $p < 0.001$) as well as across recording modes (S1[with model] × S2[without model]: $\rho_{19} = 0.59$, $p = 0.003$; S1[without model] × S2[with model]: $\rho_{19} = 0.63$, $p = 0.001$). Comparisons of the two within-modes sentence correlations with each of the two across-modes sentence correlations did not indicate significant differences (all $0.153 \leq z \leq 1.704$; all $0.088 \leq ps \leq 0.879$).

[5]In response to a reviewer request, we performed the analogous analyses on sentence F0 (cf. also electronic supplementary material, table S20), as determined across the entire sample and with the same settings for pitch extraction (see above). The Pearson correlation between the sum and the difference of sentence F0 in the two recording modes ($r_{19} = -0.18$, $p = 0.436$) did not significantly differ from zero, $t_{19} = -0.798$, $p = 0.435$ (two-tailed).

# 4. General discussion

## 4.1. Relationships between attractiveness and distinctiveness

The present study is to our knowledge, the first to demonstrate a systematic relationship between perceived attractiveness and distinctiveness in human voices. Here, we found strong negative correlations between attractiveness and deviation-based distinctiveness (DEV) for voices when based both on vowels ($\rho = -0.85$) and on sentences ($\rho = -0.87$). This pattern was analogous to and, if anything, even stronger than the previously described negative correlation between attractiveness and DEV for faces (experiment 2: $\rho = -0.64$; experiment 4: $\rho = -0.74$). Overall, this pattern of findings provides strong support for an averageness account of attractiveness for both faces and voices [3,7] when distinctiveness is assessed in a deviation-based manner. Note, that the negative relationship between attractiveness and DEV was not merely an artefact of imitating a model speaker during voice recordings: in experiment 5 using new speakers, we replicated our results for voices recorded with the model, but also found significant, though smaller, negative correlations for voices recorded without model speaker (vowels: $\rho = -0.46$, and sentences: $\rho = -0.59$). While this indicates that the presence of a model partially preserves idiosyncratic variability in voices which drives the relationship between attractiveness and DEV in terms of stable 'voice traits', speaking after a model enhances the strength of this relationship, perhaps owing to a change of natural voice variation affecting either attractiveness, DEV or both. Based on correlations across recording modes, which were substantial and positive for attractiveness, but tended to be relatively smaller for DEV ratings (at least for vowels), we tentatively suggest that the presence of a model speaker may change natural variation of DEV more than variation of attractiveness. Note, however, that mean F0 was remarkably similar across the recording modes. The notion that the relationship between attractiveness and DEV is systematic and substantial, independent of recording mode, is further supported by strong and positive cross-sentence correlations throughout for attractiveness ratings ($0.64 \leq \rho \leq 0.77$) as well as for DEV ratings ($0.63 \leq \rho \leq 0.79$), with no significant difference between recording modes.

The present findings are important because they were obtained in the context of ratings for real stimuli, rather than for averaged 'composite' stimuli. Note that averaging towards composites causes artefacts *per se*, such as smooth and symmetric visual patterns for faces, or increased harmonics-to-noise ratios for voices. (It should also be noted, however, that voice morphing is a comparatively new and elaborate technique which only a few laboratories master, and this may also explain why there are relatively few studies investigating the averageness account of vocal attractiveness.) We propose that, because such 'non-average' features of digitally created composites have been shown to consistently contribute to perceived attractiveness [7,14], studies with natural individual stimuli that vary in perceived prototypicality or averageness are important to cross-validate findings obtained with composite stimuli.

Compared to these consistent findings of negative correlations between attractiveness and deviation-based distinctiveness, the relationship with rated attractiveness was much less consistent for 'in-the-crowd'-based distinctiveness ratings. Specifically, while there was also a moderate negative correlation between attractiveness and 'in-the-crowd'-based distinctiveness (VITC) for voices when based on vowels (experiment 1), this correlation was positive when based on sentences (experiment 3). For faces, a marginally non-significant positive correlation between attractiveness and FITC was found in experiment 1 (numerically similar to the significant positive correlation with more stimuli as reported in [34]), and while this pattern was not seen in experiment 3 using the same stimuli and task, the relationships between rated attractiveness and DITC ratings were inconsistent when compared to DEV ratings. Taken together, our findings confirm that common deviation-based and 'in-the-crowd'-based measures of distinctiveness (VITC and FITC) measure at least partially different constructs [34], and extend those findings by showing that this is the case both for faces and for voices. For voices, the specific relationship between attractiveness and distinctiveness appears to depend on the type of utterance. Specifically, while simple vowel stimuli were rated as less attractive with increasing VITC distinctiveness (experiment 1), in line with an averageness account, sentence stimuli (experiment 3) were rated as more attractive with increasing VITC distinctiveness.

These differences between different types of utterances may generally be related to differences in duration and/or number of different phonemes [6,59]. Whereas sentences carry much richer cues to attractiveness and distinctiveness, vowels are simple periodic utterances which are mainly influenced by 'static' biophysical characteristics of individual speakers. VITC ratings for vowels could also differ from those for sentences owing to a certain oddity of imagining someone saying a prolonged vowel

sound in a crowd. Possibly related to this notion, an earlier study reported that perceived voice attractiveness and acoustic distance to mean (in terms of F0, F1) were correlated for a vowel (/a/), but not for a word or sentence [6]. Overall, ratings of attractiveness and VITC distinctiveness are probably based on partially different sets of acoustic cues, depending on their salience in a given utterance. The positive correlation between voice attractiveness and VITC distinctiveness is reminiscent of analogous findings for faces in previous research [34] where it has been argued that DITC measures of distinctiveness may be distorted by cognitive heuristics. Accordingly, raters might be biased to think that they would surely spot a highly attractive person in the crowd, even when this might not be the case. Such an effect seems to generalize to voices in the present study, at least when ratings are based on sentence stimuli. Note that such a putative heuristic, as suggested here, does not imply that attractive voices would, in fact, stand out of a crowd if put to test. (In fact, other more salient bottom-up acoustic characteristics such as intensity [60] probably play a more prominent role here which we had controlled in our stimuli by RMS normalization.) While we are at present unaware of studies addressing the specific issue of whether attractive voices stick out from a noisy environment, a recent study on the 'cocktail-party effect' [61] could provide tentative and indirect evidence in favour of this assumption. Specifically, interference from a non-target speaker can be reduced both when the target is familiar and the interfering voice is unfamiliar and, critically, also when the target is unfamiliar and the interfering voice is familiar [62]. Although the link to attractiveness is indirect, voice familiarity, just like voice averaging (and, by implication, attractiveness), could promote positive evaluation via a fluency mechanism as seen in the mere exposure effect.

In contrast to 'in-the-crowd-based' measures of distinctiveness, correlations for deviation-based distinctiveness and attractiveness were highly consistent, and consistently negative across modalities and utterance types in the present study. This supports the notion that both faces and voices become increasingly attractive the more typical, i.e. the more average, they are perceived relative to prior personal experience [3,7]. At variance with this experience-based account of typicality, it has been argued recently that typicality ratings rather reflect stereotypes of what constitutes attractive and typical voices [9]. In our view, this may be the case for tasks that do not further specify what typicality/distinctiveness is. However, it should be noted that our task explicitly invoked a memory component by asking participants to judge distinctiveness relative to the faces and voices they know. Given the different patterns of results for two types of distinctiveness measures, we believe that it is extremely important for future studies to specify exactly how typicality/distinctiveness was assessed.

Overall, while deviation-based measures gave rise to a highly consistent pattern of negative correlations with attractiveness across stimulus modalities and domains, inconsistent correlations were seen for attractiveness and DITC measures which may be distorted by subjective heuristics. This may indicate that DEV ratings are preferable over 'in-the-crowd'-based measures to assess distinctiveness in an unbiased manner.

## 4.2. Relationships between ratings for faces and voices of the same speakers

The second aim of the present study was to provide a systematic assessment of relationships between ratings of attractiveness and distinctiveness for faces and voices from the same speakers. Positive correlations between independent ratings of faces and voices might be expected to the extent that (i) facial and vocal features are determined by the same underlying basis (e.g. genetic or hormonal), and (ii) those features systematically influence perceptions under investigation (e.g. of attractiveness or distinctiveness). A common basis of vocal and facial attractiveness has been postulated by several studies (e.g. [35,41,43]). While we are unaware of research directly linking attractiveness and hormonal status via distinctiveness, there is evidence that certain vocal parameters (e.g. F0, vocal tract length estimates, shimmer, jitter, harmonics-to-noise ratio, as determined from sustained vowel recordings only) are linked to speakers' body size measurements (e.g. height, weight and waist-to-hip ratios), as probably mediated by hormonal mechanisms [63]. The present findings, however, consistently indicate that correlations between vocal and facial attractiveness are remarkably absent in the majority of the studied conditions, and small at best in one exception which we discuss below (figure 2). It could be argued that the standardization of the present stimuli in terms of neutral emotional expression and speaking style according to a model speaker, may have compromised to some degree the natural variation between voices relevant for perceived attractiveness, such as vocal pitch.

However, experiment 5 addresses this concern, and its results are clear in showing that correlations between facial and vocal attractiveness in sentences were also absent for voices recorded naturally and without a model speaker, as predicted [55] based on our findings from experiment 4. Although we

had no predictions regarding the small positive correlation ($\rho = 0.30$) between facial and vocal attractiveness for simple vowels we had observed in experiment 1, it may be noted that this was not replicated in experiment 5. Rather, we found a numerically negative, though non-significant correlation ($\rho = -0.30$) in the condition with model speaker, and a numerically negative non-significant correlation ($\rho = -0.18$) in the new condition without a model speaker. Overall, the pattern of results across five experiments would seem to indicate that, for a range of conditions tested in this series of experiments, any correlation between facial and vocal attractiveness is small at best, and is potentially non-existent.

Together, the present findings challenge the 'honest signal account' of facial and vocal attractiveness [14]. On one hand, we appreciate that the only exception to this pattern, a small but significant positive correlation between facial and vocal attractiveness in experiment 1, when simple vowels were used as voice samples, could potentially resolve discrepancies between our data and previous findings in which evidence for a correlation between facial and vocal attractiveness was reported using similarly simple vocalizations [41,43]. On the other hand, our failure to replicate this finding with a new set of speakers emphasizes the importance for researchers both to critically consider the nature of the stimuli used to assess these relationships, and to assess the replicability of critical findings across a range of conditions and situations.

In that respect, prerequisites to find evidence for or against the honest signal hypothesis include that face and voice stimuli should be honest and undistorted representations of their owners' genetic quality. We selected our face stimuli to be devoid of attractiveness-enhancing features such as make-up or jewellery. However, it may be more difficult to remove or standardize socio-cultural norms of attractiveness that are reflected in acquired speech patterns in the voice [9]. In that sense, the present voice ratings to more naturalistic sentence stimuli may in part reflect cultural norms, rather than purely biophysically determined voice qualities. Accordingly, one explanation for the results found with simple vowels could be that these are relatively devoid of such socio-cultural cues, and thus may reflect genetic factors more 'honestly' compared to more naturalistic and complex vocalizations. While this interesting possibility should be addressed in more detail in future research, we can conclude that correlations between vocal and facial attractiveness appear to be remarkably absent, at least when voices are presented in the more naturalistic context of sentences (as opposed to vowels) akin to everyday communication.

Although acoustic analyses on vowel pitch and sentence duration did not indicate different degrees of acoustic variability for samples recorded with, versus without, the model speaker, electronic supplementary material, table S19 suggest approximately 10% longer average durations of the same sentences when produced with compared to without a model speaker. We tentatively attribute this difference to the larger effort to imitate neutral emotion and speaking style of a model.

With respect to distinctiveness, facial and vocal ratings were uncorrelated for both types of distinctiveness ratings, suggesting no common basis for perceived distinctiveness. An interesting question for future research is how various measures of vocal distinctiveness could be related to one another. For instance, it would be instructive to assess in more detail how DITC and DEV are related with the actual recognizability of voices (for relevant research on faces, see [34,64,65]), and to determine the acoustic stimulus parameters which underlie different aspects of perceived vocal distinctiveness (for relevant methods, see [66]).

## 4.3. Limitations

As a possible limitation for both the present study and earlier research in this field [35,40,41,43], we assessed attractiveness and distinctiveness for static faces, and thus did not consider a possible role of dynamic facial information. To the extent that static and dynamic faces may be judged by different standards [67], it remains possible that cross-domain correlations between facial and vocal attractiveness could be found for dynamic facial stimuli. In fact, one previous study emphasized the role of dynamic information for correlations between vocal and visual attractiveness, although this was not found consistently across different conditions [68]. Recent theoretical accounts of person perception increasingly address the role of dynamic information [69], and this issue warrants further investigation.

As a second step towards understanding impression formation in every-day social interaction, it may be of interest how faces and voices combine to shape our evaluation of a person's attractiveness. Clearly, simultaneous presentation of face-voice stimuli would be unsuited to study the honest-signal account of attractiveness which requires independent ratings of (unimodal) voices and faces, owing to possible multimodal interactions. Interestingly, such interactions present a promising research field in their own

right, as they can reveal important insights into the relative contribution of facial and vocal information to social evaluations beyond attractiveness (see [70,71]).

## 5. Conclusion

This study establishes, for the first time to our knowledge, substantial and systematic differences between deviation-based and 'in-the-crowd'-based measures of voice distinctiveness and their relationship to perceived attractiveness. Specifically, we found strong negative correlations between DEV and attractiveness, both for voices and faces, such that more distinctive speakers were perceived as less attractive. This supports the averageness account of attractiveness for both domains. By contrast, we found no such negative correlations between DITC and attractiveness, instead, correlations between DITC and attractiveness were marginally positive for faces, and significantly positive for voices when speakers uttered sentences, such that more distinctive voices were rated as more attractive. These findings strongly suggests that both measures of distinctiveness reflect at least partially different constructs, a finding which needs to be considered in future studies on vocal and facial distinctiveness.

An important finding of this research was that vocal and facial attractiveness ratings for the same speakers were remarkably uncorrelated, as were vocal and facial distinctiveness ratings. Overall, these findings challenge the honest signal hypothesis of facial and vocal attractiveness at least in the context of ratings for relatively naturalistic, i.e. sentence-based, speech and emphasize the importance to consider in detail the nature of the stimuli used for this type of research.

Ethics. This study was approved by the Faculty Ethics Committee of the University of Jena (permit no. FSV 14/02). All speakers gave their written informed consent to the use of their faces and voices as stimuli in the current and future experiments. All participants gave their written informed consent to participate in the study.

Data accessibility. Our data [72] are deposited at the Open Science Framework (OSF) platform: https://osf.io/cef43/ (doi:10.31234/osf.io/2avu3)

Authors' contributions. R.Z. designed the study, analysed the data, interpreted the results and wrote the manuscript. S.R.S. and V.G.S. contributed to designing the study, interpreting the data and writing the manuscript. All authors gave final approval for this publication.

Competing interests. The authors declare no competing interests.

Funding. This work was supported by the Deutsche Forschungsgemeinschaft (DFG; ZA 745/1–1 and 745/1-2, granted to R.Z.; and FOR1097, granted to S.R.S.).

Acknowledgements. We thank the Deutsche Forschungsgemeinschaft (DFG) for financial support of this study. We cordially thank Constanze Mühl, Marie-Christin Perlich, Denise Humble, Mathias Riedel, Rebecca Wientzek, Charlotte Bargou, Jessica Munstein and Benito Romanelli for help with stimulus recording, editing and data acquisition.

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
