## [Reviewer comments · Royal Society Open Science]

Review History

RSOS-181177.R0 (Original submission)

Review form: Reviewer 1

Is the manuscript scientifically sound in its present form?

No

Are the interpretations and conclusions justified by the results?

No

Is the language acceptable?

Yes

Is it clear how to access all supporting data?

No

Do you have any ethical concerns with this paper?

No

Have you any concerns about statistical analyses in this paper?

Yes

Recommendation?

Reject

Comments to the Author(s)

The authors investigated relationships between rated distinctiveness and rated vocal or facial attractiveness in a small sample of men and women. Although the general research question of multi-modality in person perception is interesting and important, and averageness is generally understudied (particularly in voice perception), the authors do not present a compelling rationale nor a clear set of hypotheses and predictions in this study. The introduction is broad and poorly grounded in theory, with several errors in logic as described below. Critically, there are a few fairly major issues with the study design (e.g., imitation in voice recording; attractiveness judgments conflated with perceived pleasantness) and with statistical analysis. By using simple bivariate correlations, the authors do not test for effects of apparent interest, such as multi-modality in attractiveness or the influence of speech type (vowels vs. phrases) on vocal ratings, nor possible interactions among these factors, and between sexes, and do not control for multiple comparisons despite conducting a very large number of spearman rank correlations.

Taken together, while the paper has the potential to offer novel insight into how people associate facial or vocal distinctiveness with attractiveness, in its current form the paper suffers from a lack of a clear theoretical basis from which to make predictions, potentially confounding methods that are likely to have influenced the outcomes of rating tasks (and their comparability with previous work), and problematic statistical analyses that make it difficult to ascertain the true pattern of results.

Below I expand on these overarching issues in more detail along with several minor comments:

Introduction

The rationale behind the author's arguments and predictions, loosely laid out in the introduction, is unclear in several instances. Critically, the authors do not provide a mechanistic nor functional account of facial attractiveness, particularly as it relates to distinctiveness or averageness – why from an evolutionary or social perspective might less distinctive faces or voices be more, or less, attractive? In many cases, there also appear to be flaws in the authors' logic, or perhaps critical elements of their argument have been omitted. For instance, while the authors very briefly note the role of hormones on determining facial and vocal sexual dimorphism (and thus the possible role of hormones in mediating covariation between facial and vocal attractiveness via facial masculinity), the link between hormones and attractiveness via facial distinctiveness is not mentioned. If there are studies to suggest that such a link exists, this literature should be cited. Similarly, the authors suggest that FITC (face-in-the-crowd) ratings should positively predict vocal attractiveness, as they often do facial attractiveness, because raters may assume that a highly attractive face/voice will surely be distinctive. Yet the same logic does not necessarily generalize from faces to voices here. Consider, for instance, that an attractive voice will not always "stick out" in a crowd of other voices given the complexities of the well-known "cocktail party effect" (which the authors do not mention). In fact there is no a priori reason to assume that attractive voices will typically be easier to hear, or that participants will make this assumption when rating the attractiveness of distinctive voices. Attractiveness and amplitude often correlate negatively, and high-pitched voices (unattractive in men) are likely to propagate across space at further distances than low-pitched voices.

Methods

Overall the experimental procedure is rigorous and clearly described, however there are some major issues with instructions: the authors describe that "speakers were asked to produce vowels

and sentences in similar style, intensity and timing, and in emotionally neutral intonation, as exemplified in a pre-recorded model speaker (first author, presented via loudspeakers) (P6, L48).” This type of voice imitation task is not typically used in voice attractiveness research, most likely because it can greatly reduce the amount of natural variation in vocal production and could therefore effectively mask the important between-individual differences in vocal parameters that contribute to attractiveness (e.g., voice pitch). Thus, it is difficult to say whether the effects (or lack thereof) reported here for ratings of voice attractiveness and distinctiveness are not simply an artefact of this unusual imitation procedure.

In addition, participants were instructed to rate attractiveness “in the sense of sounding pleasant”, which can also produce different ratings than when no further instructions (beyond attractiveness) are given. This too may have influenced the results and makes it difficult to compare these results with those of previous studies on voice/face attractiveness, where “pleasantness” (P8 L48) is rarely ever noted.

References should be provided for the instructions given to participants, including for VITC and DEV ratings (P8-9). Were these statements taken from previous work, or generated by the authors?

How was sample size determined?

Results

For the research question of interest, spearman rank correlations provide an over simplistic way to examine relationships between voice and face attractiveness. Simple correlations do not allow the authors to examine the effects of ‘speech type’ and ‘modality (face/voice)’, as well as other potentially relevant variables such as sex of stimulus, sex of rater, and utterance duration (for voice stimuli). These factors may very well interact and if so, this could entirely change the interpretation of any simple correlations.

Thus, I propose that the authors run an omnibus multi-factorial model such as a linear mixed model. Simple correlations can be provided to supplement an omnibus model, for instance in a supplementary table. However, with such a large number of correlations, the authors really should control for multiple comparisons using corrective algorithms such as Bonferroni or the less conservative Sidak. The authors’ justification for not using such a group-wise correction (“we likely would have missed small but important effects, such as small positive correlations between voice and face attractiveness in possible support of the honest signal hypothesis”, P9), probably won’t satisfy most researchers. The problem is that we cannot know whether these “small but important effects” really do offer support for the authors’ hypothesis, or are simply a statistical artefact, as suggested for example by some of the contradictory results for which the authors had no apparent a priori predictions (e.g., negative correlations for vowels and positive for sentences).

Minor comments

As the experimental design is within-subjects, it is unusual to refer to this study as comprising 4 separate experiments, rather than 4 experimental conditions.

2.2 Raters – Can the authors clarify why handedness is relevant to report?

P9 L31 – The authors should note here (rather than on page 10) how familiarity judgments were collected, and whether this bimodal judgment was provided for each voice/face on each trial.

P4 L16 – Provide references for FITC and DEV measures.

P4 L28 – Note how attractiveness was quantified here (I assume rated on a scale).

P4 L46 – Another recent review on the topic is Groyecka et al. (2017)

P5 L4 – The authors have omitted a large number of studies that have shown positive correlations between facial and vocal attractiveness in one or both sexes (e.g., Hughes & Miller, 2015; Little et al., 2011; Saxton, Burriss, et al., 2009; Skrinda et al., 2014; Wheatley et al., 2014)

P9 L22+ Untypical should read atypical

P11 L12 – Here, and in the abstract, nonsignificant relationships should not be noted as “marginally positive” but rather “marginally nonsignificant” (especially at a $P > .10$, even before correcting for multiple comparisons).

There are missing numbers in the sequential numbering of supplementary Tables (e.g., Tables S4, S5, S7, S9, S11, S13, S15 and S17 are missing from the supplemental document yet some of these Table numbers are references in the paper, whereas many supplementary tables are not referenced at all)

References

Groyecka, A. et al. (2017) Attractiveness Is multimodal: beauty Is also in the nose and ear of the beholder. *Front. Psychol.* 8:778. doi: 10.3389/fpsyg.2017.00778

Hughes, S. M. & Miller, N. E. (2015). What sounds beautiful looks beautiful stereotype: the matching of attractiveness of voices and faces. *Journal of Social and Personal Relationships*, 33(7), 1– 14.

Little, A. C., Connely, J., Feinberg, D. R., Jones, B. C., & Roberts, S. C. (2011). Human preference for masculinity differs according to context in faces, bodies, voices, and smell. *Behavioral Ecology*, 22(4), 862– 868. [http:// doi.org/ 10.1093/ beheco/ arr061](http://doi.org/10.1093/beheco/arr061)

Pisanski, K. & Feinberg, D. (2018). Voice attractiveness. *The Oxford Handbook of Voice Perception*.

Puts, D.A., Hill, A.K., Bailey, D.H., Walker, R.S., Rendall, D., Wheatley, J.R., Welling, L.L., Dawood, K., Cárdenas, R., Burriss, R.P. and Jablonski, N.G. (2016). Sexual selection on male vocal fundamental frequency in humans and other anthropoids. *Proc. R. Soc. B*, 283(1829):20152830.

Saxton, T. K., Debruine, L. M., Jones, B. C., Little, A. C., & Roberts, S. C. (2009). Face and voice attractiveness judgments change during adolescence. *Evolution and Human Behavior*, 30(6), 398– 408. [http:// doi.org/ 10.1016/ j.evolhumbehav.2009.06.004](http://doi.org/10.1016/j.evolhumbehav.2009.06.004)

Skrinda, I., Krama, T., Kecko, S., Moore, F. R., Kaasik, A., Meija, L., . . . Krams, I. (2014). Body height, immunity, facial and vocal attractiveness in young men. *Naturwissenschaften*, 101(12), 1017– 1025. [http:// doi.org/ 10.1007/ s00114- 014- 1241- 8](http://doi.org/10.1007/s00114-014-1241-8)

Wheatley, J. R., Apicella, C. A., Burriss, R. P., Cárdenas, R. A., Bailey, D. H., Welling, L. L. M., & Puts, D. A. (2014). Women’s faces and voices are cues to reproductive potential in industrial and forager societies. *Evolution and Human Behavior*, 35(4), 264– 271. [http:// doi.org/ 10.1016/ j.evolhumbehav.2014.02.006](http://doi.org/10.1016/j.evolhumbehav.2014.02.006)

Review form: Reviewer 2

Is the manuscript scientifically sound in its present form?

Yes

Are the interpretations and conclusions justified by the results?

No

Is the language acceptable?

Yes

Is it clear how to access all supporting data?

No

Do you have any ethical concerns with this paper?

No

Have you any concerns about statistical analyses in this paper?

No

Recommendation?

Accept with minor revision (please list in comments)

Comments to the Author(s)

The paper by Zaeske et al reports four experiments that assess the relationship of face and voice attractiveness as well as the relationship between attractiveness and different types of distinctiveness. The conclusions are that voice and face attractiveness are uncorrelated despite a small but significant correlation between attractiveness of static faces and vowels and a small but near-significant correlation in a second experiment.

The paper is well written. It addresses interesting questions, which will make an important contribution to the field. I do recommend publication after the following two points are considered:

- (1) The correlations seem to be carried out on raw data. That is unusual because different participants will vary in how they use the 6-point scale (some participants will hover more in the middle - others will use the entire scale). It is therefore recommended to z-score or min-max transform the data. I wonder why the authors did not do that.
- (2) I find the conclusions too "radical". The fairest/most appropriate auditory equivalent to a static face is a non-semantically meaningful, short utterance such as a vowel sound. I am therefore not surprised that the correlations between "static face attractiveness" and "speech attractiveness" are not significant. In my view, you are comparing two unmatched signals. Speech will inevitably contain many more variables, which will elicit some kind of positive or negative person judgment. These additional variables will interact in interesting - and so far largely unexplored - ways with (in this case) attractiveness. The information you are adding to the voice carrying speech that is not apparent from a short vowel sound or a static face image are, for example, accents as signal to social class, speech velocity (often related to perceived intelligence), idiosyncrasies in pronunciation of words (e.g. a lisp) etc. All of these will affect attractiveness ratings and are no longer a "pure" reflection. Related to this point are TWO small correlations at .3 and .24 (one significant and one nearly significant at $p = .059$) between attractiveness ratings of static face and short vowel stimuli. This is despite using the raw data and despite only small variation in attractiveness ratings (neither extremes seem well represented). Thus I find the title and the conclusions too harsh and would recommend a revision of both.

Decision letter (RSOS-181177.R0)

18-Sep-2018

Dear Dr Zäske:

Manuscript ID RSOS-181177 entitled "Attractiveness and distinctiveness between speakers"

voices and faces are uncorrelated" which you submitted to Royal Society Open Science, has been reviewed. The comments from reviewers are included at the bottom of this letter.

In view of the criticisms of the reviewers, the manuscript has been rejected in its current form. However, a new manuscript may be submitted which takes into consideration these comments.

Please note that resubmitting your manuscript does not guarantee eventual acceptance, and that your resubmission will be subject to peer review before a decision is made.

Your resubmitted manuscript should be submitted by 18-Mar-2019. If you are unable to submit by this date please contact the Editorial Office.

Please note that Royal Society Open Science will introduce article processing charges for all new submissions received from 1 January 2018. Charges will also apply to papers transferred to Royal Society Open Science from other Royal Society Publishing journals, as well as papers submitted as part of our collaboration with the Royal Society of Chemistry (<http://rsos.royalsocietypublishing.org/chemistry>). If your manuscript is submitted and accepted for publication after 1 Jan 2018, you will be asked to pay the article processing charge, unless you request a waiver and this is approved by Royal Society Publishing. You can find out more about the charges at <http://rsos.royalsocietypublishing.org/page/charges>. Should you have any queries, please contact openscience@royalsociety.org.

on behalf of Dr Carolyn McGettigan (Associate Editor) and Prof. Antonia Hamilton (Subject Editor)
openscience@royalsociety.org

Reviewers' Comments to Author:
Reviewer: 1

Comments to the Author(s)

The authors investigated relationships between rated distinctiveness and rated vocal or facial attractiveness in a small sample of men and women. Although the general research question of multi-modality in person perception is interesting and important, and averageness is generally understudied (particularly in voice perception), the authors do not present a compelling rationale nor a clear set of hypotheses and predictions in this study. The introduction is broad and poorly grounded in theory, with several errors in logic as described below. Critically, there are a few fairly major issues with the study design (e.g., imitation in voice recording; attractiveness judgments conflated with perceived pleasantness) and with statistical analysis. By using simple bivariate correlations, the authors do not test for effects of apparent interest, such as multi-modality in attractiveness or the influence of speech type (vowels vs. phrases) on vocal ratings,

nor possible interactions among these factors, and between sexes, and do not control for multiple comparisons despite conducting a very large number of spearman rank correlations.

Taken together, while the paper has the potential to offer novel insight into how people associate facial or vocal distinctiveness with attractiveness, in its current form the paper suffers from a lack of a clear theoretical basis from which to make predictions, potentially confounding methods that are likely to have influenced the outcomes of rating tasks (and their comparability with previous work), and problematic statistical analyses that make it difficult to ascertain the true pattern of results.

Below I expand on these overarching issues in more detail along with several minor comments:

Introduction

The rationale behind the author's arguments and predictions, loosely laid out in the introduction, is unclear in several instances. Critically, the authors do not provide a mechanistic nor functional account of facial attractiveness, particularly as it relates to distinctiveness or averageness – why from an evolutionary or social perspective might less distinctive faces or voices be more, or less, attractive? In many cases, there also appear to be flaws in the authors' logic, or perhaps critical elements of their argument have been omitted. For instance, while the authors very briefly note the role of hormones on determining facial and vocal sexual dimorphism (and thus the possible role of hormones in mediating covariation between facial and vocal attractiveness via facial masculinity), the link between hormones and attractiveness via facial distinctiveness is not mentioned. If there are studies to suggest that such a link exists, this literature should be cited. Similarly, the authors suggest that FITC (face-in-the-crowd) ratings should positively predict vocal attractiveness, as they often do facial attractiveness, because raters may assume that a highly attractive face/voice will surely be distinctive. Yet the same logic does not necessarily generalize from faces to voices here. Consider, for instance, that an attractive voice will not always “stick out” in a crowd of other voices given the complexities of the well-known “cocktail party effect” (which the authors do not mention). In fact there is no a priori reason to assume that attractive voices will typically be easier to hear, or that participants will make this assumption when rating the attractiveness of distinctive voices. Attractiveness and amplitude often correlate negatively, and high-pitched voices (unattractive in men) are likely to propagate across space at further distances than low-pitched voices.

Methods

Overall the experimental procedure is rigorous and clearly described, however there are some major issues with instructions: the authors describe that “speakers were asked to produce vowels and sentences in similar style, intensity and timing, and in emotionally neutral intonation, as exemplified in a pre-recorded model speaker (first author, presented via loudspeakers) (P6, L48).” This type of voice imitation task is not typically used in voice attractiveness research, most likely because it can greatly reduce the amount of natural variation in vocal production and could therefore effectively mask the important between-individual differences in vocal parameters that contribute to attractiveness (e.g., voice pitch). Thus, it is difficult to say whether the effects (or lack thereof) reported here for ratings of voice attractiveness and distinctiveness are not simply an artefact of this unusual imitation procedure.

In addition, participants were instructed to rate attractiveness “in the sense of sounding pleasant”, which can also produce different ratings than when no further instructions (beyond attractiveness) are given. This too may have influenced the results and makes it difficult to compare these results with those of previous studies on voice/face attractiveness, where “pleasantness” (P8 L48) is rarely ever noted.

References should be provided for the instructions given to participants, including for VITC and DEV ratings (P8-9). Were these statements taken from previous work, or generated by the authors?

How was sample size determined?

Results

For the research question of interest, spearman rank correlations provide an over simplistic way to examine relationships between voice and face attractiveness. Simple correlations do not allow the authors to examine the effects of ‘speech type’ and ‘modality (face/voice)’, as well as other potentially relevant variables such as sex of stimulus, sex of rater, and utterance duration (for voice stimuli). These factors may very well interact and if so, this could entirely change the interpretation of any simple correlations.

Thus, I propose that the authors run an omnibus multi-factorial model such as a linear mixed model. Simple correlations can be provided to supplement an omnibus model, for instance in a supplementary table. However, with such a large number of correlations, the authors really should control for multiple comparisons using corrective algorithms such as Bonferroni or the less conservative Sidak. The authors’ justification for not using such a group-wise correction (“we likely would have missed small but important effects, such as small positive correlations between voice and face attractiveness in possible support of the honest signal hypothesis”, P9), probably won’t satisfy most researchers. The problem is that we cannot know whether these “small but important effects” really do offer support for the authors’ hypothesis, or are simply a statistical artefact, as suggested for example by some of the contradictory results for which the authors had no apparent a priori predictions (e.g., negative correlations for vowels and positive for sentences).

Minor comments

As the experimental design is within-subjects, it is unusual to refer to this study as comprising 4 separate experiments, rather than 4 experimental conditions.

2.2 Raters – Can the authors clarify why handedness is relevant to report?

P9 L31 – The authors should note here (rather than on page 10) how familiarity judgments were collected, and whether this bimodal judgment was provided for each voice/face on each trial.

P4 L16 – Provide references for FITC and DEV measures.

P4 L28 – Note how attractiveness was quantified here (I assume rated on a scale).

P4 L46 – Another recent review on the topic is Groyecka et al. (2017)

P5 L4 – The authors have omitted a large number of studies that have shown positive correlations between facial and vocal attractiveness in one or both sexes (e.g., Hughes & Miller, 2015; Little et al., 2011; Saxton, Burriss, et al., 2009; Skrinda et al., 2014; Wheatley et al., 2014)

P9 L22+ Untypical should read atypical

P11 L12 – Here, and in the abstract, nonsignificant relationships should not be noted as “marginally positive” but rather “marginally nonsignificant” (especially at a $P > .10$, even before correcting for multiple comparisons).

There are missing numbers in the sequential numbering of supplementary Tables (e.g., Tables S4, S5, S7, S9, S11, S13, S15 and S17 are missing from the supplemental document yet some of these Table numbers are references in the paper, whereas many supplementary tables are not referenced at all)

References

- Groyecka, A. et al. (2017) Attractiveness Is multimodal: beauty Is also in the nose and ear of the beholder. *Front. Psychol.* 8:778. doi: 10.3389/fpsyg.2017.00778
- Hughes, S. M. & Miller, N. E. (2015). What sounds beautiful looks beautiful stereotype: the matching of attractiveness of voices and faces. *Journal of Social and Personal Relationships*, 33(7), 1- 14.
- Little, A. C., Connely, J., Feinberg, D. R., Jones, B. C., & Roberts, S. C. (2011). Human preference for masculinity differs according to context in faces, bodies, voices, and smell. *Behavioral Ecology*, 22(4), 862– 868. [http:// doi.org/ 10.1093/ beheco/ arr061](http://doi.org/10.1093/beheco/arr061)
- Pisanski, K. & Feinberg, D. (2018). Voice attractiveness. *The Oxford Handbook of Voice Perception*.
- Puts, D.A., Hill, A.K., Bailey, D.H., Walker, R.S., Rendall, D., Wheatley, J.R., Welling, L.L., Dawood, K., Cárdenas, R., Burriss, R.P. and Jablonski, N.G. (2016). Sexual selection on male vocal fundamental frequency in humans and other anthropoids. *Proc. R. Soc. B*, 283(1829):20152830.
- Saxton, T. K., Debruine, L. M., Jones, B. C., Little, A. C., & Roberts, S. C. (2009). Face and voice attractiveness judgments change during adolescence. *Evolution and Human Behavior*, 30(6), 398– 408. [http:// doi.org/ 10.1016/ j.evolhumbehav.2009.06.004](http://doi.org/10.1016/j.evolhumbehav.2009.06.004)
- Skrimda, I., Krama, T., Kecko, S., Moore, F. R., Kaasik, A., Meija, L., . . . Krams, I. (2014). Body height, immunity, facial and vocal attractiveness in young men. *Naturwissenschaften*, 101(12), 1017– 1025. [http:// doi.org/ 10.1007/ s00114- 014- 1241- 8](http://doi.org/10.1007/s00114-014-1241-8)
- Wheatley, J. R., Apicella, C. A., Burriss, R. P., Cárdenas, R. A., Bailey, D. H., Welling, L. L. M., & Puts, D. A. (2014). Women’s faces and voices are cues to reproductive potential in industrial and forager societies. *Evolution and Human Behavior*, 35(4), 264– 271. [http:// doi.org/ 10.1016/ j.evolhumbehav.2014.02.006](http://doi.org/10.1016/j.evolhumbehav.2014.02.006)

Reviewer: 2

Comments to the Author(s)

The paper by Zaeske et al reports four experiments that assess the relationship of face and voice attractiveness as well as the relationship between attractiveness and different types of distinctiveness. The conclusions are that voice and face attractiveness are uncorrelated despite a small but significant correlation between attractiveness of static faces and vowels and a small but near-significant correlation in a second experiment.

The paper is well written. It addresses interesting questions, which will make an important contribution to the field. I do recommend publication after the following two points are considered:

- (1) The correlations seem to be carried out on raw data. That is unusual because different participants will vary in how they use the 6-point scale (some participants will hover more in the middle - others will use the entire scale). It is therefore recommended to z-score or min-max transform the data. I wonder why the authors did not do that.
- (2) I find the conclusions too "radical". The fairest/most appropriate auditory equivalent to a static face is a non-semantically meaningful, short utterance such as a vowel sound. I am therefore not surprised that the correlations between "static face attractiveness" and "speech attractiveness" are not significant. In my view, you are comparing two unmatched signals. Speech will inevitably contain many more variables, which will elicit some kind of positive or negative person judgment. These additional variables will interact in interesting - and so far

largely unexplored – ways with (in this case) attractiveness. The information you are adding to the voice carrying speech that is not apparent from a short vowel sound or a static face image are, for example, accents as signal to social class, speech velocity (often related to perceived intelligence), idiosyncrasies in pronunciation of words (e.g. a lisp) etc. All of these will affect attractiveness ratings and are no longer a “pure” reflection.

Related to this point are TWO small correlations at .3 and .24 (one significant and one nearly significant at $p = .059$) between attractiveness ratings of static face and short vowel stimuli. This is despite using the raw data and despite only small variation in attractiveness ratings (neither extremes seem well represented). Thus I find the title and the conclusions to harsh and would recommend a revision of both.

Author's Response to Decision Letter for (RSOS-181177.R0)

See Appendices A & B.

RSOS-190429.R0

Review form: Reviewer 1

Is the manuscript scientifically sound in its present form?

Yes

Are the interpretations and conclusions justified by the results?

No

Is the language acceptable?

Yes

Is it clear how to access all supporting data?

Yes

Do you have any ethical concerns with this paper?

No

Have you any concerns about statistical analyses in this paper?

Yes

Recommendation?

Major revision is needed (please make suggestions in comments)

Comments to the Author(s)

Thank you to the authors for providing detailed responses to my comments and attempting to address all issues that arose in the initial review of this paper. Most initial comments have been adequately addressed through the addition and revision of text. In particular, the authors have nicely clarified the theoretical rationale and predictions of their study in the introduction and have included relevant supporting references that were initially omitted. They have also briefly noted a critical limitation (model speaker, see revised ms document pg18 L 35) and future directions (multi-modal tests, pg32 L18) in their revised discussion. The paper reads quite nicely. However, while I appreciate the justifications provided by the authors in their response to

referees, three of the most critical issues (related to methodology and analysis) have not, in my opinion, been effectively addressed.

These outstanding issues include:

1. Model voice and imitation

The issue still remains that speakers in the study were first played a 'model voice' and then asked to "produce vowels and sentences naturally but in emotionally neutral intonation, and in similar style, intensity, and timing [as the model voice]" (revised ms, page 20). As noted in my original review, this type of imitation task will surely affect (alter) the natural or baseline properties of a person's voice, as the speaker may attempt to imitate the style but also, whether wittingly or not, the frequency patterns of the model voice. This could, very likely, be enough to override natural individual differences in these nonverbal vocal parameters that are (in their baseline state) predictive of various mate-relevant traits that listeners may rely on when judging vocal attractiveness.

In their response to referees (and in the new text now added to the revised ms, pg 20), the reviewers argue that this method was used "To reduce influences of momentary voice „states“ (such as a specific emotional expression or speaking style), and to ensure that between-speaker variation was instead characteristic for more robust individual voice „traits“. In my opinion, imitating a model voice has quite the opposite effect and rather washes away between-individual differences.

In their response to referees, the authors acknowledge that it "may be difficult to categorically exclude the possibility that this procedure might have caused a small degree of reduction of relevant acoustic variability in the voice samples, we believe that it is unlikely that our effects (or lack thereof) are an artefact of this procedure. This is because our sentence data (cf. Fig. 1, Exp. 2 and 4) show a substantial degree of variability in vocal attractiveness between speakers, and vocal attractiveness ratings also show a highly systematic pattern of large negative correlations with distinctiveness. (this is roughly also noted also in new discussion of results in revised ms pg 30 L 35).

However, a high degree of variability in vocal attractiveness ratings between-speakers, or covariation between attractiveness and distinctiveness, could be due to a number of factors elicited by the voice imitation task. For instance, some people sound quite "odd" (clearly not speaking in their natural voice) when attempting to imitate a model voice. So, in theory, the attractiveness ratings could be tracking how "natural" vs "weird" a speaker sounds. In short, the use of a model voice is a real limitation of the study because it likely to have affected the vocal production of speakers in a way that would also affect listeners' ratings of the speakers' vocal attractiveness. In other words, it confounds the main research question of this particular study.

2. 'Attractiveness' vs 'pleasantness'

The authors asked participants to rate faces based on how "good looking" they are, and to rate voices based on how "pleasant" they sound. In their response to referees (as in the new text added to the revised ms), the authors argue that, "we assumed that asking for attractiveness without further specification would potentially evoke several different attractiveness concepts across raters (e.g. sexual attractiveness, likeability, aesthetics, beauty). We therefore aimed to disambiguate the term by providing a synonym which we reasoned would be more readily understood."

I completely agree with the authors that offering a working definition can help to disambiguate results, especially in rating tasks, however not when two different definitions are provided for face (good looking) and voice (pleasantness) judgments, which are then treated as the same measure. Attractiveness and pleasantness are quite clearly two different constructs. A young male, for instance, can imagine the voice of an elderly woman as being pleasant (reminiscent perhaps of his beloved grandmother) yet not attractive.

Similar to the issue raised in point 1, the terminology used in the rating tasks for faces versus voices could have easily affected listeners' ratings and thus, the relationship (or lack

thereof) between face and voice attractiveness ratings could be due to either, or both, of these methodological aspects of the study.

3. Simple, uncorrected correlations

As noted earlier, an omnibus model (e.g., LMM or even an ANOVA) would allow the authors to directly test for main and interaction effects of stimulus type (vowels vs. phrases) and other relevant factors. Linear models would also allow additional control of effects of speaker ID across conditions. However, the authors have opted to maintain their simple analysis of a series of spearman ranked correlations, without controlling for multiple comparisons. In their response the authors argue that, "we decided to not correct for multiple correlations to avoid false negative conclusions". Yet without doing so, the authors invite false positives.

At the very minimum, the authors should control for multiple comparisons in **all** of their correlations (rather than a select few, as now presented in supplement, because 'choosing' which correlations to correct defeats the purpose). A more sensible solution to the issue of false negatives would be to increase sample sizes, as N=10 participants per sex, per experiment, is a very small sample size.

Review form: Reviewer 2

Is the manuscript scientifically sound in its present form?

Yes

Are the interpretations and conclusions justified by the results?

Yes

Is the language acceptable?

Yes

Is it clear how to access all supporting data?

Not Applicable

Do you have any ethical concerns with this paper?

No

Have you any concerns about statistical analyses in this paper?

No

Recommendation?

Accept as is

Comments to the Author(s)

I have no further comments.

Decision letter (RSOS-190429.R0)

17-Apr-2019

Dear Dr Zäske:

Manuscript ID RSOS-190429 entitled "Attractiveness and distinctiveness between speakers"

voices in naturalistic speech and their faces are uncorrelated" which you submitted to Royal Society Open Science, has been reviewed. The comments from reviewer(s) are included at the bottom of this letter.

In view of the criticisms of the reviewer(s), I must decline the manuscript for publication in Royal Society Open Science at this time. However, a new manuscript may be submitted which takes into consideration these comments.

Please note that resubmitting your manuscript does not guarantee eventual acceptance, and that your resubmission will be subject to re-review by the reviewer(s) before a decision is rendered.

You will be unable to make your revisions on the originally submitted version of your manuscript. Instead, revise your manuscript using a word processing program and save it on your computer.

You may also click the below link to start the resubmission process (or continue the process if you have already started your resubmission) for your manuscript. If you use the below link you will not be required to login to ScholarOne Manuscripts.

*** PLEASE NOTE: This is a two-step process. After clicking on the link, you will be directed to a webpage to confirm. ***

https://mc.manuscriptcentral.com/rsos?URL_MASK=ee60cb3986d6487e82ab3fc4dcd5ff51

Because we are trying to facilitate timely publication of manuscripts submitted to Royal Society Open Science, your resubmitted manuscript should be submitted by 15-Oct-2019. If you are unable to submit by this date please contact the Editorial Office for options.

I look forward to a resubmission.

on behalf of Professor Carolyn McGettigan (Associate Editor) and Professor Antonia Hamilton (Subject Editor)
openscience@royalsociety.org

Associate Editor Comments to Author (Professor Carolyn McGettigan):

We have now received comments on your revised manuscript from the two original reviewers. You will see that they are both very pleased with several aspects of the revision. However, one reviewer re-iterates three critical issues on which they are not yet satisfied. Having read the revised manuscript in detail, I tend to agree with their evaluation. Points 1 and 2, to my mind, cannot be fully addressed to the reviewer's satisfaction without running a new study. On point 3, I share the reviewer's opinion that the results should at the least be reported with the sufficient level of correction for multiple comparisons, and the authors have already expressed in their previous response that they do not wish to follow this particular recommendation. Taking all of this into account, I am unfortunately recommending that the paper is rejected at this point. It may be that adding a follow-up experiment using a different method to obtain voice recordings (i.e.

not imitations), as well as more controlled definitions of attractiveness, would help to resolve some of the issues to allow for publication of this work in the future. I will leave the option open for a new submission, should the authors decide to collect new data.

Reviewer comments to Author:

Reviewer: 2

Comments to the Author(s)

I have no further comments.

Reviewer: 1

Comments to the Author(s)

Thank you to the authors for providing detailed responses to my comments and attempting to address all issues that arose in the initial review of this paper. Most initial comments have been adequately addressed through the addition and revision of text. In particular, the authors have nicely clarified the theoretical rationale and predictions of their study in the introduction and have included relevant supporting references that were initially omitted. They have also briefly noted a critical limitation (model speaker, see revised ms document pg18 L 35) and future directions (multi-modal tests, pg32 L18) in their revised discussion. The paper reads quite nicely. However, while I appreciate the justifications provided by the authors in their response to referees, three of the most critical issues (related to methodology and analysis) have not, in my opinion, been effectively addressed.

These outstanding issues include:

1. Model voice and imitation

The issue still remains that speakers in the study were first played a 'model voice' and then asked to "produce vowels and sentences naturally but in emotionally neutral intonation, and in similar style, intensity, and timing [as the model voice]" (revised ms, page 20). As noted in my original review, this type of imitation task will surely affect (alter) the natural or baseline properties of a person's voice, as the speaker may attempt to imitate the style but also, whether wittingly or not, the frequency patterns of the model voice. This could, very likely, be enough to override natural individual differences in these nonverbal vocal parameters that are (in their baseline state) predictive of various mate-relevant traits that listeners may rely on when judging vocal attractiveness.

In their response to referees (and in the new text now added to the revised ms, pg 20), the reviewers argue that this method was used "To reduce influences of momentary voice „states“ (such as a specific emotional expression or speaking style), and to ensure that between-speaker variation was instead characteristic for more robust individual voice „traits“. In my opinion, imitating a model voice has quite the opposite effect and rather washes away between-individual differences.

In their response to referees, the authors acknowledge that it "may be difficult to categorically exclude the possibility that this procedure might have caused a small degree of reduction of relevant acoustic variability in the voice samples, we believe that it is unlikely that our effects (or lack thereof) are an artefact of this procedure. This is because our sentence data (cf. Fig. 1, Exp. 2 and 4) show a substantial degree of variability in vocal attractiveness between speakers, and vocal attractiveness ratings also show a highly systematic pattern of large negative correlations with distinctiveness. (this is roughly also noted also in new discussion of results in revised ms pg 30 L 35).

However, a high degree of variability in vocal attractiveness ratings between-speakers, or covariation between attractiveness and distinctiveness, could be due to a number of factors elicited by the voice imitation task. For instance, some people sound quite "odd" (clearly not speaking in their natural voice) when attempting to imitate a model voice. So, in theory, the attractiveness ratings could be tracking how "natural" vs "weird" a speaker sounds.

In short, the use of a model voice is a real limitation of the study because it likely to have affected the vocal production of speakers in a way that would also affect listeners' ratings of the speakers'

vocal attractiveness. In other words, it confounds the main research question of this particular study.

2. 'Attractiveness' vs 'pleasantness'

The authors asked participants to rate faces based on how "good looking" they are, and to rate voices based on how "pleasant" they sound. In their response to referees (as in the new text added to the revised ms), the authors argue that, "we assumed that asking for attractiveness without further specification would potentially evoke several different attractiveness concepts across raters (e.g. sexual attractiveness, likeability, aesthetics, beauty). We therefore aimed to disambiguate the term by providing a synonym which we reasoned would be more readily understood."

I completely agree with the authors that offering a working definition can help to disambiguate results, especially in rating tasks, however not when two different definitions are provided for face (good looking) and voice (pleasantness) judgments, which are then treated as the same measure. Attractiveness and pleasantness are quite clearly two different constructs. A young male, for instance, can imagine the voice of an elderly woman as being pleasant (reminiscent perhaps of his beloved grandmother) yet not attractive.

Similar to the issue raised in point 1, the terminology used in the rating tasks for faces versus voices could have easily affected listeners' ratings and thus, the relationship (or lack thereof) between face and voice attractiveness ratings could be due to either, or both, of these methodological aspects of the study.

3. Simple, uncorrected correlations

As noted earlier, an omnibus model (e.g., LMM or even an ANOVA) would allow the authors to directly test for main and interaction effects of stimulus type (vowels vs. phrases) and other relevant factors. Linear models would also allow additional control of effects of speaker ID across conditions. However, the authors have opted to maintain their simple analysis of a series of spearman ranked correlations, without controlling for multiple comparisons. In their response the authors argue that, "we decided to not correct for multiple correlations to avoid false negative conclusions". Yet without doing so, the authors invite false positives.

At the very minimum, the authors should control for multiple comparisons in **all** of their correlations (rather than a select few, as now presented in supplement, because 'choosing' which correlations to correct defeats the purpose). A more sensible solution to the issue of false negatives would be to increase sample sizes, as N=10 participants per sex, per experiment, is a very small sample size.

Author's Response to Decision Letter for (RSOS-190429.R0)

See Appendix C.

RSOS-201244.R0

Review form: Reviewer 1

Is the manuscript scientifically sound in its present form?

No

Are the interpretations and conclusions justified by the results?

No

Is the language acceptable?

Yes

Do you have any ethical concerns with this paper?

No

Have you any concerns about statistical analyses in this paper?

Yes

Recommendation?

Major revision is needed (please make suggestions in comments)

Comments to the Author(s)

The authors have now included new data (Exp. 5) based on a small sample of 21 vocalizers and 20 listeners, to address some of my earlier outstanding concerns. I thank the authors for this effort. Unfortunately, the predictions that the authors choose to test and report (or not) in this new experiment do not directly address nor counter-act the issues that still persist in the earlier data (Exp 1-4), as I describe below. Moreover, while I find that the first part of the paper (Exp 1-4) remains well written and structured, with clear methods and procedures, the new Exp. 5 is not well integrated into the manuscript and reads more like an appendix.

Most importantly my initial concerns regarding the statistical tests have not been addressed in this revision. The authors choose to run only simple correlations in all five experiments, without correcting for multiple comparisons, substantially weakening the reliability and interpretation of their results (including new data). It is difficult to interpret the new results in light of this and again, I strongly recommend that the authors conduct linear mixed models in which speaker ID and listener ID could be included as random factors, and variables of interest (e.g., utterance type, sex of speaker, model/no model) could be included in the LMM as fixed factors. Mixed models have more statistical power by accounting for additional noise in the data (e.g., due to between-rater biases / preferences). Pairwise comparisons could then be conducted to explore significant effects revealed by the models.

There remains no correction for multiple pairwise comparisons. The authors' justification for this is not convincing:

- For Exp 1-4: in the manuscript the authors argue "We considered that a problem of using an alpha adjustment for multiple tests (such as the Bonferroni correction) could have been that we likely would have missed small but important effects, such as small positive correlations between voice and face attractiveness in possible support of the honest signal hypothesis" (p 14). In their response to referees they further note that "We appreciate that false positive results due to multiple comparisons can be a valid and important concern. In the present study, we would have thought that this is less of a concern, to the extent that systematic patterns of findings were obtained in multiple independent data sets." But this is not true. The authors find only one, weak relationship between voice and face ratings in Exp. 1-4 ($\rho = .30$, $p = .014$). In Exp 5 this result is not replicated, and in fact, the opposite relationship is found (a negative correlation). Thus, this single weak relationship is not likely 'evidence in support of the honest signal hypothesis' but more likely to be a false positive.

- For Exp. 5: the authors further argue "We did not correct for multiple tests since tests refer to pre-registered hypotheses." (p 17). This is not a valid justification, as even a priori predicted relationships are susceptible to false positives.

Given the small sample sizes in all five experiments and inability to replicate the only significant correlation, I encourage the authors to impose a correction for all spearman correlations, within the main text/results of the paper. I suspect that many of the weaker/moderate 'significant' relationships reported are not likely to survive.

New data: Experiment 5

I thank the authors for conducting a new experiment to address two major issues. First, that 'naturalistic' voice stimuli used in Exp 1-4 are not really naturalistic (sentences were read,

emotionally neutral, and speakers were played a model voice and tasked with imitating the style, intensity and timing), and second that 'attractiveness' was defined for faces as 'good looking' and for voices as 'pleasant'.

As noted above it is difficult to interpret the new results of Exp 5 on the basis of oversimplified, uncorrected correlation tests, and in the absence of several other tests that would more effectively support Exp. 1-4 methods. Thus there remain problems with the initial Exps 1-4 that Exp 5 does not directly overcome.

First, in Exp. 5, raters were asked to rate 'attractiveness' (without any context) for both voices and faces. This addresses the difference in terminology between the modes. But the results of Exp 5 still do not tell us if 'pleasantness' ratings are reliable indices of voice attractiveness, nor show that these ratings can be directly compared with previous research on voice attractiveness, as the authors claim. "Attractiveness" and "pleasantness" are largely independent constructs, where a voice could indeed be judged low on attractiveness yet still be perceived as highly pleasant. The authors argue in Exp 1-4 that "By adding the synonyms "goodlooking" and "sounding pleasant", we reasoned that participants would refer to a similar concept that would instantly make sense for the respective modality" (pg 10), but they give no justification or empirical evidence for this assumption.

Second, in Exp. 5, speakers still read the sentences but they did not hear a 'model' voice. This addresses the major problem of imitation. Nevertheless, it is important to note that the speech stimuli are still not 'naturalistic' (i.e., read, not 'free' speech).

Importantly, to test whether acoustic variability is higher for samples recorded without vs. with a model speaker, the authors compare mean F0 and Fo-SD measured from two vowels, rather than from the sentence stimuli (Table S18). Differences in F0 and F0-SD between the model/no model conditions would be expected in the sentences, not the vowels.

It is also not clear how F0 and F0-SD were measured (more details regarding acoustic analysis should be given).

Finally, the authors compare attractiveness ratings between model/no model sentence stimuli but do not make the more relevant comparison in distinctiveness (DEV) ratings between the two recording modes.

A remaining theoretical weakness of the paper, that could be addressed in the intro and discussion, stems from a lack of strong rationale/evidence linking voice averageness with attractiveness. The authors rightfully note in their paper that sexual dimorphism is typically attractive in voices (as in faces), as supported by dozens of empirical studies. In contrast the arguments (and empirical evidence) for voice averageness as attractive remain less convincing. Indeed, only a couple of studies (those cited in the paper, ref 7 and 8) have examined voice averageness, and these studies are now a decade old. That research remains controversial for a number of reasons including no clear rationale for predicting that voice averageness should be attractive.

Review form: Reviewer 2

Is the manuscript scientifically sound in its present form?

Yes

Are the interpretations and conclusions justified by the results?

Yes

Is the language acceptable?

Yes

Do you have any ethical concerns with this paper?

No

Have you any concerns about statistical analyses in this paper?

No

Recommendation?

Accept as is

Comments to the Author(s)

A lovely paper!

Decision letter (RSOS-201244.R0)

Dear Dr Zäske

The Editors assigned to your paper RSOS-201244 "Attractiveness and distinctiveness between speakers' voices in naturalistic speech and their faces are uncorrelated" have now received comments from reviewers and would like you to revise the paper in accordance with the reviewer comments and any comments from the Editors. Please note this decision does not guarantee eventual acceptance.

Please submit your revised manuscript and required files (see below) no later than 21 days from today's (ie 10-Aug-2020) date. Note: the ScholarOne system will 'lock' if submission of the revision is attempted 21 or more days after the deadline. If you do not think you will be able to meet this deadline please contact the editorial office immediately.

on behalf of Prof Essi Viding (Subject Editor)
 openscience@royalsociety.org

Reviewer comments to Author:
 Reviewer: 2

Comments to the Author(s)
 A lovely paper!

Reviewer: 1

Comments to the Author(s)

The authors have now included new data (Exp. 5) based on a small sample of 21 vocalizers and 20 listeners, to address some of my earlier outstanding concerns. I thank the authors for this effort. Unfortunately, the predictions that the authors choose to test and report (or not) in this new experiment do not directly address nor counter-act the issues that still persist in the earlier data (Exp 1-4), as I describe below. Moreover, while I find that the first part of the paper (Exp 1-4) remains well written and structured, with clear methods and procedures, the new Exp. 5 is not well integrated into the manuscript and reads more like an appendix.

Most importantly my initial concerns regarding the statistical tests have not been addressed in this revision. The authors choose to run only simple correlations in all five experiments, without correcting for multiple comparisons, substantially weakening the reliability and interpretation of their results (including new data). It is difficult to interpret the new results in light of this and again, I strongly recommend that the authors conduct linear mixed models in which speaker ID and listener ID could be included as random factors, and variables of interest (e.g., utterance type, sex of speaker, model/no model) could be included in the LMM as fixed factors. Mixed models have more statistical power by accounting for additional noise in the data (e.g., due to between-rater biases / preferences). Pairwise comparisons could then be conducted to explore significant effects revealed by the models.

There remains no correction for multiple pairwise comparisons. The authors' justification for this is not convincing:

- For Exp 1-4: in the manuscript the authors argue "We considered that a problem of using an alpha adjustment for multiple tests (such as the Bonferroni correction) could have been that we likely would have missed small but important effects, such as small positive correlations between voice and face attractiveness in possible support of the honest signal hypothesis" (p 14). In their response to referees they further note that "We appreciate that false positive results due to multiple comparisons can be a valid and important concern. In the present study, we would have thought that this is less of a concern, to the extent that systematic patterns of findings were obtained in multiple independent data sets." But this is not true. The authors find only one, weak relationship between voice and face ratings in Exp. 1-4 ($\rho = .30$, $p = .014$). In Exp 5 this result is not replicated, and in fact, the opposite relationship is found (a negative correlation). Thus, this single weak relationship is not likely 'evidence in support of the honest signal hypothesis' but more likely to be a false positive.

- For Exp. 5: the authors further argue “We did not correct for multiple tests since tests refer to pre-registered hypotheses.” (p 17). This is not a valid justification, as even a priori predicted relationships are susceptible to false positives.

Given the small sample sizes in all five experiments and inability to replicate the only significant correlation, I encourage the authors to impose a correction for all spearman correlations, within the main text/results of the paper. I suspect that many of the weaker/moderate ‘significant’ relationships reported are not likely to survive.

New data: Experiment 5

I thank the authors for conducting a new experiment to address two major issues. First, that ‘naturalistic’ voice stimuli used in Exp 1-4 are not really naturalistic (sentences were read, emotionally neutral, and speakers were played a model voice and tasked with imitating the style, intensity and timing), and second that ‘attractiveness’ was defined for faces as ‘good looking’ and for voices as ‘pleasant’.

As noted above it is difficult to interpret the new results of Exp 5 on the basis of oversimplified, uncorrected correlation tests, and in the absence of several other tests that would more effectively support Exp. 1-4 methods. Thus there remain problems with the initial Exps 1-4 that Exp 5 does not directly overcome.

First, in Exp. 5, raters were asked to rate ‘attractiveness’ (without any context) for both voices and faces. This addresses the difference in terminology between the modes. But the results of Exp 5 still do not tell us if ‘pleasantness’ ratings are reliable indices of voice attractiveness, nor show that these ratings can be directly compared with previous research on voice attractiveness, as the authors claim. “Attractiveness” and “pleasantness” are largely independent constructs, where a voice could indeed be judged low on attractiveness yet still be perceived as highly pleasant. The authors argue in Exp 1-4 that “By adding the synonyms “goodlooking” and “sounding pleasant”, we reasoned that participants would refer to a similar concept that would instantly make sense for the respective modality” (pg 10), but they give no justification or empirical evidence for this assumption.

Second, in Exp. 5, speakers still read the sentences but they did not hear a ‘model’ voice. This addresses the major problem of imitation. Nevertheless, it is important to note that the speech stimuli are still not ‘naturalistic’ (i.e., read, not ‘free’ speech).

Importantly, to test whether acoustic variability is higher for samples recorded without vs. with a model speaker, the authors compare mean F0 and Fo-SD measured from two vowels, rather than from the sentence stimuli (Table S18). Differences in F0 and F0-SD between the model/no model conditions would be expected in the sentences, not the vowels.

It is also not clear how F0 and F0-SD were measured (more details regarding acoustic analysis should be given).

Finally, the authors compare attractiveness ratings between model/no model sentence stimuli but do not make the more relevant comparison in distinctiveness (DEV) ratings between the two recording modes.

A remaining theoretical weakness of the paper, that could be addressed in the intro and discussion, stems from a lack of strong rationale/evidence linking voice averageness with attractiveness. The authors rightfully note in their paper that sexual dimorphism is typically attractive in voices (as in faces), as supported by dozens of empirical studies. In contrast the arguments (and empirical evidence) for voice averageness as attractive remain less convincing. Indeed, only a couple of studies (those cited in the paper, ref 7 and 8) have examined voice averageness, and these studies are now a decade old. That research remains controversial for a number of reasons including no clear rationale for predicting that voice averageness should be attractive.

===PREPARING YOUR MANUSCRIPT===

===PREPARING YOUR REVISION IN SCHOLARONE===

<https://royalsociety.org/journals/authors/author-guidelines/#supplementary-material> to include a suitable title and informative caption. An example of appropriate titling and captioning may be found at https://figshare.com/articles/Table_S2_from_Is_there_a_trade-off_between_peak_performance_and_performance_breadth_across_temperatures_for_aerobic_sc_ope_in_teleost_fishes_/3843624.

Author's Response to Decision Letter for (RSOS-201244.R0)

See Appendix D.

RSOS-201244.R1 (Revision)

Review form: Reviewer 1

Is the manuscript scientifically sound in its present form?

Yes

Are the interpretations and conclusions justified by the results?

Yes

Is the language acceptable?

Yes

Do you have any ethical concerns with this paper?

No

Have you any concerns about statistical analyses in this paper?

Yes

Recommendation?

Accept with minor revision (please list in comments)

Comments to the Author(s)

I thank the authors for their detailed responses to my concerns and for running additional tests, including these as footnotes / exploratory analyses. While I feel that the methods and results retain some issues, that authors have now made reference to these limitations in the paper, thus leaving the judgment to the reader.

Following my earlier comment that has not been fully addressed, I recommend that the authors include stronger rationale (a few sentences) in the Introduction to describe why we might predict average voices to be attractive (from an evolutionary, social, perceptual or any other point of view). While the lack of empirical evidence in the literature linking voice attractiveness and averageness may indeed be tied to methodological obstacles of voice resynthesis, this does not negate the importance of having an a priori prediction as to why such a relationship should exist - and why that is being investigated in the current set of studies.

The new Table S20 (comparison of F0 measured from sentences produced with/without a model) is missing from the supplementary file.

Wishing you all well in these hard times.

Decision letter (RSOS-201244.R1)

Dear Dr Zäske

On behalf of the Editors, we are pleased to inform you that your Manuscript RSOS-201244.R1 "Attractiveness and distinctiveness between speakers' voices in naturalistic speech and their faces are uncorrelated" has been accepted for publication in Royal Society Open Science subject to minor revision in accordance with the referees' reports. Please find the referees' comments along with any feedback from the Editors below my signature.

Please submit your revised manuscript and required files (see below) no later than 7 days from today's (ie 12-Nov-2020) date. Note: the ScholarOne system will 'lock' if submission of the revision is attempted 7 or more days after the deadline. If you do not think you will be able to meet this deadline please contact the editorial office immediately.

Please note article processing charges apply to papers accepted for publication in Royal Society Open Science (<https://royalsocietypublishing.org/rsos/charges>). Charges will also apply to papers transferred to the journal from other Royal Society Publishing journals, as well as papers

submitted as part of our collaboration with the Royal Society of Chemistry (<https://royalsocietypublishing.org/rsos/chemistry>). Fee waivers are available but must be requested when you submit your revision (<https://royalsocietypublishing.org/rsos/waivers>).

on behalf of the Associate Editor and Professor Essi Viding (Subject Editor)
openscience@royalsociety.org

Associate Editor Comments to Author:

Thank you for submitting your revised manuscript. Please address Reviewer #1's remaining comments and submit your new revised paper with a point-by-point response.

Reviewer comments to Author:

Reviewer: 1
Comments to the Author(s)

I thank the authors for their detailed responses to my concerns and for running additional tests, including these as footnotes / exploratory analyses. While I feel that the methods and results retain some issues, that authors have now made reference to these limitations in the paper, thus leaving the judgment to the reader.

Following my earlier comment that has not been fully addressed, I recommend that the authors include stronger rationale (a few sentences) in the Introduction to describe why we might predict average voices to be attractive (from an evolutionary, social, perceptual or any other point of view). While the lack of empirical evidence in the literature linking voice attractiveness and averageness may indeed be tied to methodological obstacles of voice resynthesis, this does not negate the importance of having an a priori prediction as to why such a relationship should exist - and why that is being investigated in the current set of studies.

The new Table S20 (comparison of F0 measured from sentences produced with/without a model) is missing from the supplementary file.

Wishing you all well in these hard times.

===PREPARING YOUR MANUSCRIPT===

===PREPARING YOUR REVISION IN SCHOLARONE===

- Ensure that your data access statement meets the requirements at <https://royalsociety.org/journals/authors/author-guidelines/#data>. You should ensure that you cite the dataset in your reference list. If you have deposited data etc in the Dryad repository, please only include the 'For publication' link at this stage. You should remove the 'For review' link.
- If you are requesting an article processing charge waiver, you must select the relevant waiver option (if requesting a discretionary waiver, the form should have been uploaded at Step 3 'File upload' above).
- If you have uploaded ESM files, please ensure you follow the guidance at <https://royalsociety.org/journals/authors/author-guidelines/#supplementary-material> to include a suitable title and informative caption. An example of appropriate titling and captioning may be found at https://figshare.com/articles/Table_S2_from_Is_there_a_trade-off_between_peak_performance_and_performance_breadth_across_temperatures_for_aerobic_sc_ope_in_teleost_fishes_/3843624.

Author's Response to Decision Letter for (RSOS-201244.R1)

See Appendix D.

Decision letter (RSOS-201244.R2)

Dear Dr Zäske,

It is a pleasure to accept your manuscript entitled "Attractiveness and distinctiveness between speakers' voices in naturalistic speech and their faces are uncorrelated" in its current form for publication in Royal Society Open Science.

on behalf of the Associate Editor, and Professor Essi Viding (Subject Editor)
openscience@royalsociety.org

Appendix A

**FRIEDRICH-SCHILLER-
UNIVERSITÄT
JENA** Institut für Psychologie
Lehrstuhl für Allgemeine Psychologie

Universität Jena · Lehrstuhl für Allgemeine Psychologie · Institut für Psychologie · 07737 Jena

Dr. Romi Zäske

Am Steiger 3/Haus 1
07743 Jena

Telefon: +49 (0) 36 41 9-45935

Telefax: +49 (0) 36 41 9-45182

E-Mail: romi.zaeske@uni-jena.de

<http://www.allgpsy.uni-jena.de/>

March 7th, 2019

To Dr Carolyn McGettigan (Associate Editor)
and Prof Antonia Hamilton (Subject Editor) of **Royal Society Open Science**

Dear Dr. McGettigan, dear Prof. Hamilton

thank you very much for inviting a resubmission of our manuscript entitled “Attractiveness and distinctiveness in voices and faces of young adults” by myself, Verena G. Skuk, and Stefan R. Schweinberger for possible publication as a *Research Article* in *Royal Society Open Science*.

Please find attached the revised version of the manuscript, carefully taking into consideration all of the reviewers’ comments of the previous round, as highlighted for convenience and detailed in the response letter. We have paid particular attention to concerns of Reviewer 1 in that we further clarified the rationale of our study, choice of task and analyses, and took into account additional literature as suggested by the reviewer. Note that in addition to the changes specified in the response letter, we have also performed some minor edits in the interest of streamlining.

We hope you agree with us, that the manuscript is now in good shape for publication in *Royal Society Open Science*. We look forward to your response.

Kind regards,

Romi Zäske, Verena Skuk, and Stefan R. Schweinberger

Response to Reviewers' Comments: Manuscript ID: RSOS-181177

Reviewers' Comments to Author: Reviewer: 1

Comments to the Author(s) The authors investigated relationships between rated distinctiveness and rated vocal or facial attractiveness in a small sample of men and women. Although the general research question of multi-modality in person perception is interesting and important, and averageness is generally understudied (particularly in voice perception), the authors do not present a compelling rationale nor a clear set of hypotheses and predictions in this study. The introduction is broad and poorly grounded in theory, with several errors in logic as described below. Critically, there are a few fairly major issues with the study design (e.g., imitation in voice recording; attractiveness judgments conflated with perceived pleasantness) and with statistical analysis. By using simple bivariate correlations, the authors do not test for effects of apparent interest, such as multi-modality in attractiveness or the influence of speech type (vowels vs. phrases) on vocal ratings, nor possible interactions among these factors, and between sexes, and do not control for multiple comparisons despite conducting a very large number of spearman rank correlations.

Taken together, while the paper has the potential to offer novel insight into how people associate facial or vocal distinctiveness with attractiveness, in its current form the paper suffers from a lack of a clear theoretical basis from which to make predictions, potentially confounding methods that are likely to have influenced the outcomes of rating tasks (and their comparability with previous work), and problematic statistical analyses that make it difficult to ascertain the true pattern of results.

Response: We thank the reviewer for these comments on our manuscript. As will be detailed below, we have made efforts to further enhance the description of the rationale for the present study, hoping to contribute to eliminating sources for potential confusion. For instance, while we agree that the role of multimodal signals for attractiveness perception is interesting in its own right, it needs to be noted that this issue is orthogonal to the purposes of the present study. Instead, we hold that assessing the „honest signal“ hypothesis (that claims that attractiveness signals good genes, and to the extent that this is the case, that there should be a positive correlation between perceived facial and vocal attractiveness ratings for the same person) requires to assess attractiveness in faces and voices independently in a unimodal situation, as we have done. Although impression formation from multimodal presentations of faces and voices therefore seems to be less relevant for the present study, we have added a citation to a very recent paper on that topic in the manuscript (Mileva, Tompkinson, Watt, & Burton, 2018, JEPHPP, see discussion section, p. 18). Please see our response to other points detailed below.

Below I expand on these overarching issues in more detail along with several minor comments: Introduction

The rationale behind the author's arguments and predictions, loosely laid out in the introduction, is unclear in several instances. Critically, the authors do not provide a

mechanistic nor functional account of facial attractiveness, particularly as it relates to distinctiveness or averageness – why from an evolutionary or social perspective might less distinctive faces or voices be more, or less, attractive? In many cases, there also appear to be flaws in the authors’ logic, or perhaps critical elements of their argument have been omitted. For instance, while the authors very briefly note the role of hormones on determining facial and vocal sexual dimorphism (and thus the possible role of hormones in mediating covariation between facial and vocal attractiveness via facial masculinity), the link between hormones and attractiveness via facial distinctiveness is not mentioned. If there are studies to suggest that such a link exists, this literature should be cited.

Response: This is a valid point, and we have now looked into this issue in some detail. Unfortunately, we were unable to find studies that specifically look into the link between hormones and attractiveness via distinctiveness. Specifically, two keyword searches with (one using „hormones AND attractiveness AND distinctiveness“ and the other using „hormones AND attractiveness AND typicality“) both yielded zero result. This search also revealed a remarkable absence of research into links between hormones and facial distinctiveness. Nevertheless, we looked into further potentially relevant papers, that made a link between hormones and voice-face/body covariations. As a result, we now quote a recent paper (Pisanski et al., 2016, Animal Behaviour) which found evidence that certain vocal parameters (e.g., F0, vocal tract length estimates, shimmer, jitter, harmonics-to-noise ratio, as determined from sustained vowel recordings only) explained a small amount of variability (up to 12%) in speakers body size measurements (e.g., height, weight, and waist-to-hip ratios). Although that study did not provide data on facial measurements or facial ratings, we mention this study when discussing our own small voice-face covariations for vowel recordings only.

Similarly, the authors suggest that FITC (face-in-the-crowd) ratings should positively predict vocal attractiveness, as they often do facial attractiveness, because raters may assume that a highly attractive face/voice will surely be distinctive. Yet the same logic does not necessarily generalize from faces to voices here. Consider, for instance, that an attractive voice will not always “stick out” in a crowd of other voices given the complexities of the well-known “cocktail party effect” (which the authors do not mention). In fact there is no a priori reason to assume that attractive voices will typically be easier to hear, or that participants will make this assumption when rating the attractiveness of distinctive voices. Attractiveness and amplitude often correlate negatively, and high-pitched voices (unattractive in men) are likely to propagate across space at further distances than low-pitched voices.

Response: We thank the reviewer for raising this interesting point. Please note that we have considered the possibility that attractiveness and amplitude might correlate negatively, and in fact we did control for this by RMS normalizing all voice stimuli, as stated in the methods section. We agree that there is no reason to assume that attractive faces always stick out, and also this is not what we have claimed. That said, we are grateful for the reviewer’s suggestion to look into the cocktail party effect. As far as we could tell from a literature search, no research exists that directly assesses

the effects of voice attractiveness for the cocktail party effect. However, we now cite a relevant study (p. 15) that shows that the cocktail party effect can be attenuated when the target speaker is familiar to the listener. Specifically, interference from a nontarget speaker can be reduced both when the target is familiar and the interfering voice is unfamiliar and, critically, also when the target is unfamiliar and the interfering voice is familiar (Johnsrude et al., 2013, Psychological Science). Although we concede that the link to attractiveness is indirect, it seems that voice familiarity, just like voice averaging (and, by implication, attractiveness), could promote positive evaluation via a fluency mechanism as seen in the mere exposure effect. As before, please note that we do not make any claims that attractive voices will actually „stick out“ in a crowd of voices. Rather, and in line with previous research, we simply argue that DITC ratings may be biased by heuristics, according to which raters are biased to think that they would likely spot a highly attractive person in a crowd.

Methods

Overall the experimental procedure is rigorous and clearly described, however there are some major issues with instructions: the authors describe that “speakers were asked to produce vowels and sentences in similar style, intensity and timing, and in emotionally neutral intonation, as exemplified in a pre-recorded model speaker (first author, presented via loudspeakers) (P6, L48).” This type of voice imitation task is not typically used in voice attractiveness research, most likely because it can greatly reduce the amount of natural variation in vocal production and could therefore effectively mask the important between-individual differences in vocal parameters that contribute to attractiveness (e.g., voice pitch). Thus, it is difficult to say whether the effects (or lack thereof) reported here for ratings of voice attractiveness and distinctiveness are not simply an artefact of this unusual imitation procedure.

Response: This is a sensible point to make. Please note that this procedure was not intended to reduce the amount of natural variation between speakers’ voices. In fact, the only reason for giving the example of the model speaker was that we aimed for stimuli which emphasized between-speaker variation that was characteristic for an individual voice „trait“ (rather than being characteristic for a momentary voice „state“ such as a specific emotional expression or speaking style). We apologize for not having made this sufficiently clear. In the revision, we clarified: „To reduce influences of momentary voice „states“ (such as a specific emotional expression or speaking style), and to ensure that between-speaker variation was instead characteristic for more robust individual voice „traits“, speakers were presented with a pre-recorded speaker (first author, presented via loudspeakers), and asked to produce vowels and sentences naturally but in emotionally neutral intonation, and in similar style, intensity, and timing.“ While it may be difficult to categorically exclude the possibility that this procedure might have caused a small degree of reduction of relevant acoustic variability in the voice samples, we believe that it is unlikely that our effects (or lack thereof) are an artefact of this procedure. This is because our sentence data (cf. Fig. 1, Exp. 2 and 4) show a substantial degree of variability in vocal attractiveness between speakers, and vocal attractiveness ratings also show a highly systematic pattern of large negative correlations with distinctiveness. In the revised

manuscript, we took the reviewer's point to address this issue in some detail in the discussion (see p. 18).

In addition, participants were instructed to rate attractiveness “in the sense of sounding pleasant”, which can also produce different ratings than when no further instructions (beyond attractiveness) are given. This too may have influenced the results and makes it difficult to compare these results with those of previous studies on voice/face attractiveness, where “pleasantness” (P8 L48) is rarely ever noted. References should be provided for the instructions given to participants, including for VITC and DEV ratings (P8-9). Were these statements taken from previous work, or generated by the authors? How was sample size determined?

Response: In terms of instructions we assumed that asking for attractiveness without further specification would potentially evoke several different attractiveness concepts across raters (e.g. sexual attractiveness, likeability, aesthetics, beauty). We therefore aimed to disambiguate the term by providing a synonym which we reasoned would be more readily understood. Furthermore, it seemed unusual and somewhat artificial to ask for attractiveness of an isolated stimulus such as a voice when, at least in German, attractiveness would more commonly be used to describe an entire person. However, in order to keep with the widely-established construct of attractiveness as used in the scientific literature on the one hand, and to offer a less ambiguous synonym on the other hand, we went for a hybrid instruction by specifying what we deemed an otherwise vague instruction (i.e. attractiveness in the sense of “good-looking” for faces, and “sounding pleasant” for voices). Although we acknowledge that minor semantic differences may exist between terms, we note that these may not necessarily compromise comparability across studies any more than semantic differences arising simply from translating “attractiveness” into German (as was done for convenience of our participants), or when using the same term across modalities. Specifically, “attractiveness” in vision may invoke a different connotation than in audition. Possibly to account for this latter possibility, a meta-analysis comparing subjective pleasantness ratings for stimuli across various domains (faces, objects, music, tastes) subsumed studies on rated “pleasantness”, “attractiveness”, “liking” or “beauty” (Kuhn & Gallinat, 2012, Neuroimage) under the same construct.

Instructions for the “Voice-in-the-crowd-distinctiveness” measure were worded in analogy to the “Face-in-the-crowd” measure as employed by Valentine & Endo (1992, QJEP) and Valentine & Bruce (1986a, Perception; 1986b; Canadian Journal of Psychology). The deviation-based distinctiveness measure was also borrowed from the face literature where both concepts have been compared previously (Wickham & Morris, 2003, American Journal of Psychology). We now provide more detail in the introduction and methods section as to the origin of our instructions including these references.

We apologize for not having mentioned a power analysis to determine sample size. A sample size of $N = 64$ face and voice stimuli was deemed sufficient based on power analyses and previous studies which found significant cross-modal correlations for attractiveness ratings with similar sample sizes (Abend, Pfluger, Koppensteiner,

Coquerelle, & Grammer, 2015, Evolution and Human Behavior; Collins & Missing, 2003, Animal Behavior; Valentova, Varella, Havlicek, & Kleisner, 2017, Behavioral Processes). Specifically, these studies had consistently reported positive face-voice correlations for attractiveness ratings. These were in the range of $r \sim .40 - .60$ and based on 30 - 42 faces and voices. Therefore, we deemed a sample size of 64 voices and faces overall sufficient to detect potential medium-sized ($r = \sim .40$) cross-modal correlations, even when analyzing female and male stimuli ($N = 32$ per gender) separately. This was confirmed by a power analysis using G-Power 3.1 which suggested a minimum sample size of $N = 34$, assuming an effect size of .40 (one-tailed) with an α error probability of .05 and a power of .80 (Faul, Erdfelder, Buchner, & Lang, 2009, Behavior Research Methods). This is now detailed in the methods section.

With respect to sample size of raters, we opted for four groups of 20 participants each (i.e. one group for each of four experiments). This results into 20 ratings per speaker, across which data have been collapsed prior to performing correlational analyses on our stimuli. Twenty ratings per speaker was deemed adequate, considering that we had obtained an excellent inter-rater reliability in a different study from our lab, which was based on 24 raters (Zäske, Skuk, Golle, & Schweinberger, under review): Cronbach's $\alpha = .95$ (for attractiveness ratings) and .85 (both for DEV-based and VITC distinctiveness ratings). We would be happy to include these considerations in the manuscript, based on this study under review, if deemed necessary by the editor.

Results

For the research question of interest, spearman rank correlations provide an over simplistic way to examine relationships between voice and face attractiveness. Simple correlations do not allow the authors to examine the effects of 'speech type' and 'modality (face/voice)', as well as other potentially relevant variables such as sex of stimulus, sex of rater, and utterance duration (for voice stimuli). These factors may very well interact and if so,

this could entirely change the interpretation of any simple correlations.

Thus, I propose that the authors run an omnibus multi-factorial model such as a linear mixed model. Simple correlations can be provided to supplement an omnibus model, for instance in a supplementary table. However, with such a large number of correlations, the authors really should control for multiple comparisons using corrective algorithms such as Bonferroni or the less conservative Sidak. The authors' justification for not using such a group-wise correction ("we likely would have missed small but important effects, such as small positive correlations between voice and face attractiveness in possible support of the honest signal hypothesis", P9), probably won't satisfy most researchers. The problem is that we cannot know whether these "small but important effects" really do offer support for the authors' hypothesis, or are simply a statistical artefact, as suggested for example by some of the contradictory results for which the authors had no apparent a priori predictions (e.g., negative correlations for vowels and positive for sentences).

Response: In the last paragraph of the results section, we now specify our analysis strategy in quite some detail. Although we decided to not correct for multiple correlations to avoid false negative conclusions, we clarify that our main finding of strong and negative correlations between attractiveness and DEV-based distinctiveness easily survive Bonferroni correction for six tests, with a corrected alpha level of .0083, as performed for each experiment. We provide extensive additional information in the form of supplemental tables for information to the reader. As the reviewer sensed, we considered that a problem of using an alpha adjustment for multiple tests (such as the Bonferroni correction) could have been that we likely would have missed small but important effects, such as small positive correlations between voice and face attractiveness in possible support of the honest-signal hypothesis. We now also provide more evidence to suggest that this was a real danger (and we anticipate that had we applied Bonferroni correction indiscriminately for specific effects for which hypotheses were derived from earlier published findings, this would have elicited criticism from another direction): In fact, although the small positive correlations between facial and vocal attractiveness for vowel stimuli correspond to a specific hypothesis and previous published research, note that these correlations would not have survived a Bonferroni correction (cf. supplementary material), and this is now also specified in the manuscript.

We hope that these specifications address the reviewer's concerns with respect to the issue of multiple correlations. To the best of our knowledge, and perhaps adding to concerns about accessibility to readers, a potential disadvantage of linear mixed models (compared to nonparametric Spearman rank correlations as used here) is that more distributional assumptions need to be made. We also considered that reviewer 2 did not make a similar suggestion. While we have thus opted, for now, to keep with our general data analysis strategy, we hope that we clarified the issues of analysis strategy and treatment of multiple tests satisfactorily.

Minor comments

As the experimental design is within-subjects, it is unusual to refer to this study as comprising 4 separate experiments, rather than 4 experimental conditions.

Response: We are afraid this is an obvious misunderstanding. Each of the four experiments tested a different sample of 20 participants each. We have checked the manuscript and made minor modifications, in an attempt to remove any residual ambiguities in the methods section.

2.2 Raters – Can the authors clarify why handedness is relevant to report?

Response: Yes and no - there is of course a literature linking facial symmetry to attractiveness, and there are some less consistent data that suggest that handedness may be related to facial asymmetry (e.g. Hardie et al., 2005, Laterality). That said, we assessed handedness in perceivers, and in fact we are happy to concede that we do not

think that this is specifically relevant to this report. For now, we have kept this information simply in the interest of detail and precision in describing the sample characteristic, in case some readers might find this information relevant for reasons we cannot anticipate at this stage, and noting that it does not consume much space (less than half a line). We would be happy to omit this information if the editor felt this was more appropriate.

P9 L31 – The authors should note here (rather than on page 10) how familiarity judgments were collected, and whether this bimodal judgment was provided for each voice/face on each trial.

Response: Thank you for pointing out this potential ambiguity. All rating tasks, including familiarity classifications, were presented blockwise as stated in the procedures. While block order was counterbalanced for the dimensions of interest (attractiveness and DEV/DITC), the familiarity judgments were always collected at the end of the experiment (cf. 2.2.2): “Finally, we recorded familiarity judgments [...].” We have moved this sentence to the end of the respective paragraph, so as to emphasize the fact that these ratings were obtained only after the main experiment.

P4 L16 – Provide references for FITC and DEV measures.

Response: Done.

P4 L28 – Note how attractiveness was quantified here (I assume rated on a scale).

Response: Done.

P4 L46 – Another recent review on the topic is Groyecka et al. (2017)

Response: Thank you. We now cite this relevant paper in the respective paragraph.

P5 L4 – The authors have omitted a large number of studies that have shown positive correlations between facial and vocal attractiveness in one or both sexes (e.g., Hughes & Miller, 2015; Little et al., 2011; Saxton, Burriss, et al., 2009; Skrinda et al., 2014; Wheatley et al., 2014)

Response: We decided to incorporate relevant research in the introduction (Skrinda et al., 2014; Wheatley et al., 2014). We decided to not discuss those studies which were neither designed nor suitable to investigate the honest-signal account of attractiveness and which therefore cannot provide evidence for or against it (Hughes & Miller, 2016, JSPR; Little, Connely, Feinberg, Jones, & Roberts, 2011; Saxton, DeBruine, Jones, Little, & Roberts, 2009). As we read this literature, Hughes and Miller looked into attractiveness stereotypes, but used faces and voices from different speakers, while Little et al. and Saxton et al. explored face and voice preferences by manipulating their stimuli with morphing techniques and pitch shifts. By contrast, in order to investigate whether voices and faces have a common biological source of

attractiveness, faces and voices need to be taken from the same speakers and need to represent veridical, unaltered stimuli.

P9 L22+ Untypical should read atypical

Response: Thank you, we have modified this accordingly throughout the manuscript.

P11 L12 – Here, and in the abstract, nonsignificant relationships should not be noted as “marginally positive” but rather “marginally nonsignificant” (especially at a $P > .10$, even before correcting for multiple comparisons).

Response: Thank you, done.

There are missing numbers in the sequential numbering of supplementary Tables (e.g., Tables S4, S5, S7, S9, S11, S13, S15 and S17 are missing from the supplemental document yet some of these Table numbers are references in the paper, whereas many supplementary tables are not referenced at all)

Response: We were a bit puzzled by this comment, since this should be complete, according to our records. We have therefore reattached our Supplemental Tables as is, and double-checked the manuscript. All supplemental tables (S1- S17) are referenced, sometimes in a row (e.g. S4 – S7; S8 – S11) when this was appropriate, which may be why not all individual table numbers can be retrieved via an automatic search, for instance.

References

- Groyecka, A. et al. (2017) Attractiveness Is multimodal: beauty Is also in the nose and ear of the beholder. *Front. Psychol.* 8:778. doi: 10.3389/fpsyg.2017.00778
- Hughes, S. M. & Miller, N. E. (2015). What sounds beautiful looks beautiful stereotype: the matching of attractiveness of voices and faces. *Journal of Social and Personal Relationships*, 33(7), 1– 14.
- Little, A. C., Connely, J., Feinberg, D. R., Jones, B. C., & Roberts, S. C. (2011). Human preference for masculinity differs according to context in faces, bodies, voices, and smell. *Behavioral Ecology*, 22(4), 862– 868. [http:// doi.org/ 10.1093/ beheco/ arr061](http://doi.org/10.1093/beheco/arr061)
- Pisanski, K. & Feinberg, D. (2018). Voice attractiveness. *The Oxford Handbook of Voice Perception*.
- Puts, D.A., Hill, A.K., Bailey, D.H., Walker, R.S., Rendall, D., Wheatley, J.R., Welling, L.L., Dawood, K., Cárdenas, R., Burriss, R.P. and Jablonski, N.G. (2016). Sexual selection on male vocal fundamental frequency in humans and other anthropoids. *Proc. R. Soc. B*, 283(1829):20152830.

Saxton, T. K., DeBruine, L. M., Jones, B. C., Little, A. C., & Roberts, S. C. (2009). Face and voice attractiveness judgments change during adolescence. *Evolution and Human Behavior*, 30(6), 398–408. <http://doi.org/10.1016/j.evolhumbehav.2009.06.004>

Skrinda, I., Krama, T., Kecko, S., Moore, F. R., Kaasik, A., Meija, L., . . . Krams, I. (2014). Body height, immunity, facial and vocal attractiveness in young men. *Naturwissenschaften*, 101(12), 1017–1025. <http://doi.org/10.1007/s00114-014-1241-8>

Wheatley, J. R., Apicella, C. A., Burriss, R. P., Cárdenas, R. A., Bailey, D. H., Welling, L. L. M., & Puts, D. A. (2014). Women's faces and voices are cues to reproductive potential in industrial and forager societies. *Evolution and Human Behavior*, 35(4), 264–271. <http://doi.org/10.1016/j.evolhumbehav.2014.02.006>

Reviewer: 2

Comments to the Author(s) The paper by Zaeske et al reports four experiments that assess the relationship of face and voice attractiveness as well as the relationship between attractiveness and different types of distinctiveness. The conclusions are that voice and face attractiveness are uncorrelated despite a small but significant correlation between attractiveness of static faces and vowels and a small but near-significant correlation in a second experiment.

The paper is well written. It addresses interesting questions, which will make an important contribution to the field. I do recommend publication after the following two points are considered: (1) The correlations seem to be carried out on raw data. That is unusual because different participants will vary in how they use the 6-point scale (some participants will hover more in the middle - others will use the entire scale). It is therefore recommended to z-score or min-max transform the data. I wonder why the authors did not do that.

Response: Thank you for your positive evaluation of our paper. Correlational analyses were not carried out on raw data, but on mean ratings per speaker (i.e. collapsed across raters) as stated at the beginning of the Results section. Please note that we have not computed parametric correlations but – for reasons including those mentioned in this comment – performed rank correlations (Spearman's rho) which should be robust against these concerns. We have now made this more explicit in the manuscript (see results section, first paragraph).

(2) I find the conclusions too "radical". The fairest/most appropriate auditory equivalent to a static face is a non- semantically meaningful, short utterance such as a vowel sound. I am therefore not surprised that the correlations between "static face attractiveness" and "speech attractiveness" are not significant. In my view, you are comparing two unmatched signals. Speech will inevitably contain many more variables, which will elicit some kind of positive or negative person judgment. These additional variables will interact in interesting – and so far largely unexplored – ways

with (in this case) attractiveness. The information you are adding to the voice carrying speech that is not apparent from a short vowel sound or a static face image are, for example, accents as signal to social class, speech velocity (often related to perceived intelligence), idiosyncrasies in pronunciation of words (e.g. a lisp) etc. All of these will affect attractiveness ratings and are no longer a “pure” reflection. Related to this point are TWO small correlations at .3 and .24 (one significant and one nearly significant at $p = .059$) between attractiveness ratings of static face and short vowel stimuli. This is despite using the raw data and despite only small variation in attractiveness ratings (neither extremes seem well represented). Thus I find the title and the conclusions to harsh and would recommend a revision of both.

Response: We can see this point. We do not wish to make a categorical counterargument here, but we should like to explain that we had deliberately chosen the title to be provocative. We feel that this is appropriate, for two reasons. First, in our perception, there has been a bias to take a face-voice correlation in attractiveness for granted – even when the relevant data often only include small samples and limited vocalisations (like vowel-only), limiting both generalizability and ecological validity. Second, the present data are very clear in finding zero correlations across a wide range of conditions that use naturalistic speech samples, and in conditions of appropriate statistical power. We therefore think it would be appropriate to emphasize, in the title, how the present results qualify earlier results. That having said, we have modified the title to do better justice to the small correlations between faces and vowel-based voice stimuli. „Attractiveness and distinctiveness between speakers faces and their voices in naturalistic speech are uncorrelated“.

We also should like to comment on the idea, as we understand it, that short vowel sounds are „pure“ reflections of voices, and that more complex and naturalistic speech samples may not be valid stimuli upon which attractiveness ratings should be based. We can see that it may be justifiable to make that argument – but we believe it is at least as justifiable to argue that short vowels are not very typical voice stimuli in everyday life. From a perspective of ecological validity, we hold that it is important to investigate any relationships between facial and vocal ratings with more naturalistic speech samples. We do agree with the reviewer that it would be interesting to collect further data that use facial videos (perhaps using parallel vowel versus sentence utterances), and we anticipate that this will be one line of future research to further refine the present findings.

References

Abend, P., Pfluger, L. S., Koppensteiner, M., Coquerelle, M., & Grammer, K. (2015). The sound of female shape: a redundant signal of vocal and facial attractiveness. *Evolution and Human Behavior*, 36(3), 174-181. doi:10.1016/j.evolhumbehav.2014.10.004

- Collins, S. A., & Missing, C. (2003). Vocal and visual attractiveness are related in women. *Animal Behaviour*, *65*, 997-1004. doi:10.1006/anbe.2000.1523
- Faul, F., Erdfelder, E., Buchner, A., & Lang, A. G. (2009). Statistical power analyses using G*Power 3.1: Tests for correlation and regression analyses. *Behavior Research Methods*, *41*(4), 1149-1160. doi:10.3758/brm.41.4.1149
- Hardie, S., Hancock, P., Rodway, P., Penton-Voak, I., Carson, D., & Wright, L. (2005). The enigma of facial asymmetry: Is there a gender-specific pattern of facedness? *Laterality*, *10*(4), 295-304. doi:10.1080/13576500442000094
- Hughes, S. M., & Miller, N. E. (2016). What sounds beautiful looks beautiful stereotype: The matching of attractiveness of voices and faces. *Journal of Social and Personal Relationships*, *33*(7), 984-996. doi:10.1177/0265407515612445
- Johnsrude, I. S., Mackey, A., Hakyemez, H., Alexander, E., Trang, H. P., & Carlyon, R. P. (2013). Swinging at a Cocktail Party: Voice Familiarity Aids Speech Perception in the Presence of a Competing Voice. *Psychological Science*, *24*(10), 1995-2004. doi:10.1177/0956797613482467
- Kuhn, S., & Gallinat, J. (2012). The neural correlates of subjective pleasantness. *Neuroimage*, *61*(1), 289-294. doi:10.1016/j.neuroimage.2012.02.065
- Little, A. C., Connely, J., Feinberg, D. R., Jones, B. C., & Roberts, S. C. (2011). Human preference for masculinity differs according to context in faces, bodies, voices, and smell. *Behavioral Ecology*, *22*(4), 862-868. doi:10.1093/beheco/arr061
- Mileva, M., Tompkinson, J., Watt, D., & Burton, A. M. (2018). Audiovisual Integration in Social Evaluation. *Journal of Experimental Psychology-Human Perception and Performance*, *44*(1), 128-138. doi:10.1037/xhp0000439
- Pisanski, K., Jones, B. C., Fink, B., O'Connor, J. J. M., DeBruine, L. M., Roder, S., & Feinberg, D. R. (2016). Voice parameters predict sex-specific body morphology in men and women. *Animal Behaviour*, *112*, 13-22. doi:10.1016/j.anbehav.2015.11.008
- Saxton, T. K., DeBruine, L. M., Jones, B. C., Little, A. C., & Roberts, S. C. (2009). Face and voice attractiveness judgments change during adolescence. *Evolution and Human Behavior*, *30*(6), 11.
- Skrinda, I., Krama, T., Kecko, S., Moore, F. R., Kaasik, A., Meija, L., . . . Krams, I. (2014). Body height, immunity, facial and vocal attractiveness in young men. *Naturwissenschaften*, *101*(12), 1017-1025. doi:10.1007/s00114-014-1241-8
- Valentine, T., & Bruce, V. (1986a). The effect of distinctiveness in recognizing and classifying faces. *Perception*, *15*(5), 525-535. doi:10.1068/p150525
- Valentine, T., & Bruce, V. (1986b). Recognizing Familiar Faces - the Role of Distinctiveness and Familiarity. *Canadian Journal of Psychology-Revue Canadienne de Psychologie*, *40*(3), 300-305. doi:10.1037/h0080101
- Valentine, T., & Endo, M. (1992). Towards An Exemplar Model of Face Processing - the Effects of Race and Distinctiveness. *Quarterly Journal of Experimental Psychology Section A-Human Experimental Psychology*, *44*(4), 671-703. doi:10.1080/14640749208401305
- Valentova, J. V., Varella, M. A. C., Havlicek, J., & Kleisner, K. (2017). Positive association between vocal and facial attractiveness in women but not in men: A cross-cultural study. *Behavioural Processes*, *135*, 95-100. doi:10.1016/j.beproc.2016.12.005
- Wheatley, J. R., Apicella, C. A., Burriss, R. P., Cardenas, R. A., Bailey, D. H., Welling, L. L. M., & Puts, D. A. (2014). Women's faces and voices are cues to reproductive potential in industrial and forager societies. *Evolution and Human Behavior*, *35*(4), 264-271. doi:10.1016/j.evolhumbehav.2014.02.006
- Wickham, L. H. V., & Morris, P. E. (2003). Attractiveness, distinctiveness, and recognition of faces: Attractive faces can be typical or distinctive but are not better recognized. *American Journal of Psychology*, *116*(3), 455-468. doi:10.2307/1423503
- Zäske, R., Skuk, V. G., Golle, J., & Schweinberger, S. R. (under review). The Jena Speaker Set (JESS) – A database of voice stimuli from unfamiliar young and old adult speakers.

Response to Reviewers' Comments: Manuscript ID: RSOS-190429

Title: "Attractiveness and distinctiveness between speakers' voices in naturalistic speech and their faces are uncorrelated"

Reviewer: 2

Comments to the Author(s)
I have no further comments.

Response #1: We thank the reviewer for her/his positive assessment.

Reviewer: 1

Comments to the Author(s)

Thank you to the authors for providing detailed responses to my comments and attempting to address all issues that arose in the initial review of this paper. Most initial comments have been adequately addressed through the addition and revision of text. In particular, the authors have nicely clarified the theoretical rationale and predictions of their study in the introduction and have included relevant supporting references that were initially omitted. They have also briefly noted a critical limitation (model speaker, see revised ms document pg18 L 35) and future directions (multi-modal tests, pg32 L18) in their revised discussion. The paper reads quite nicely. However, while I appreciate the justifications provided by the authors in their response to referees, three of the most critical issues (related to methodology and analysis) have not, in my opinion, been effectively addressed.

These outstanding issues include:

1. Model voice and imitation

The issue still remains that speakers in the study were first played a 'model voice' and then asked to "produce vowels and sentences naturally but in emotionally neutral intonation, and in similar style, intensity, and timing [as the model voice]" (revised ms, page 20). As noted in my original review, this type of imitation task will surely affect (alter) the natural or baseline properties of a person's voice, as the speaker may attempt to imitate the style but also, whether wittingly or not, the frequency patterns of the model voice. This could, very likely, be enough to override natural individual differences in these nonverbal vocal parameters that are (in their baseline state) predictive of various mate-relevant traits that listeners may rely on when judging vocal attractiveness.

In their response to referees (and in the new text now added to the revised ms, pg 20), the reviewers argue that this method was used "To reduce influences of momentary voice „states“ (such as a specific emotional expression or speaking style), and to ensure that between-speaker variation was instead characteristic for more robust individual voice „traits“". In my opinion, imitating a model voice has quite the opposite effect and rather washes away between-individual differences.

In their response to referees, the authors acknowledge that it "may be difficult to categorically exclude the possibility that this procedure might have caused a small degree of reduction of relevant acoustic variability in the voice samples, we believe that it is unlikely that our effects (or lack thereof) are an artefact of this procedure. This is because our sentence data (cf. Fig. 1, Exp. 2 and 4) show a substantial degree of variability in vocal attractiveness between speakers, and vocal attractiveness ratings also show a highly systematic pattern of large negative correlations with distinctiveness. (this is roughly also noted also in new discussion of results in revised ms pg 30 L 35).

However, a high degree of variability in vocal attractiveness ratings between-speakers, or covariation between attractiveness and distinctiveness, could be due to a number of factors elicited by the voice imitation task. For instance, some people sound quite "odd" (clearly not speaking in their natural voice) when attempting to imitate a model voice. So, in theory, the attractiveness ratings could be tracking how "natural" vs "weird" a speaker sounds.

In short, the use of a model voice is a real limitation of the study because it likely to have affected the vocal production of speakers in a way that would also affect listeners' ratings of the speakers' vocal attractiveness. In other words, it confounds the main research question of this particular study.

Response #1: We thank the reviewer for these thoughtful comments, and have followed the advice of controlling for the recording procedures. We have now conducted a new Experiment (Exp. 5) and report data for voice and face attractiveness and DEV ratings as well as acoustic analyses on a new set of speakers (N = 21), as rated by new participants (N = 20). Importantly, we recorded the same speech material (2 sentences and vowels from Exp. 1 – 4) in two recording modes, i.e. either with or without the model speaker (ensuring of course that the recording block without model speaker always came first). Listeners rated DEV and attractiveness for all stimuli in a within-subjects design, with otherwise identical procedures as in Exp. 1-4. The only exception was that instructions for attractiveness ratings were identical for both faces and voices (see also Response #2), omitting the specifications "good looking" and "sounding pleasant". The results supported our initial claim of substantial and negative correlations between attractiveness and DEV distinctiveness, both for stimuli recorded with and without the model. Notably, these correlations were smaller (vowels: $\rho = -.46$, and sentences: $\rho = -.59$) without the model, compared to with the model (vowels: $\rho = -.82$, sentences: $\rho = -.89$). While this supports our notion that more robust voice "traits" are preserved when introducing a model, it also argues for additional, more transient effects of the model on natural voice variability, which might enhance the relationship between attractiveness and DEV. Further support for a minor role of the model speaker in eliciting these correlations, is the finding that correlations between two different utterances (i.e. between two sentences and two vowels, respectively), were strong and independent of recording mode (with or without model). In terms of acoustic variability in voices we found no evidence for differences in voices recorded with vs. without model, neither in vowel pitch (F0) nor sentences duration. We chose these parameters respectively, because they were likely to vary in our stimulus set. We added this additional experiment to the manuscript, discuss its results, and have also included information on acoustic measurements in two additional tables (Supplementary Tables S18 and S19).

Please note, that we preregistered our study design, hypotheses and analyses of Experiment 5 on OSF (osf.io/5627f).

2. 'Attractiveness' vs 'pleasantness'

The authors asked participants to rate faces based on how "good looking" they are, and to rate voices based on how "pleasant" they sound. In their response to referees (as in the new text added to the revised ms), the authors argue that, "we assumed that asking for attractiveness without further specification would potentially evoke several different attractiveness concepts across raters (e.g. sexual attractiveness, likeability, aesthetics, beauty). We therefore aimed to disambiguate the term by providing a synonym which we reasoned would be more readily understood."

I completely agree with the authors that offering a working definition can help to disambiguate results, especially in rating tasks, however not when two different definitions are provided for face (good looking) and voice (pleasantness) judgments, which are then treated as the same measure. Attractiveness and pleasantness are quite clearly two different constructs. A young male, for instance, can imagine the voice of an elderly woman as being pleasant (reminiscent perhaps of his beloved grandmother) yet not attractive.

Similar to the issue raised in point 1, the terminology used in the rating tasks for faces versus voices could have easily affected listeners' ratings and thus, the relationship (or lack thereof) between face and voice attractiveness ratings could be due to either, or both, of these methodological aspects of the study.

Response #2: We addressed the reviewer's concerns in an additional experiment (Exp. 5), now reported in the manuscript (see also Response #2), and changed the instructions such that they would be entirely analogous for both modalities, with no further specifications, what is meant by "attractive": "Please assess how attractive/unattractive the voices/faces are." In line with our pre-registered hypotheses (osf.io/5627f) based on Exp. 1 -4, we found no evidence for a correlation

between face and voice attractiveness based on sentences. This lends further support to our argument that the honest signal account does not apply to more complex, naturalistic speech. In fact, it may be worth noting that the small positive correlation ($\rho = .30$) between facial and vocal attractiveness for simple vowels we had reported in Experiment 1 was not replicated in Experiment 5; to the contrary, we found a numerically negative, though non-significant correlation ($\rho = -.30$) in the condition with model speaker, and a numerically negative non-significant correlation ($\rho = -.18$) in the new condition without model speaker. Overall, this pattern of results across five experiments would seem to indicate that, for a range of conditions tested in this series of experiments, any correlation between facial and vocal attractiveness is small at best, and is potentially non-existent.

We added this additional experiment to the manuscript, discuss its results, and have also included information on acoustic measurements in two additional tables (Supplementary Tables S18 and S19). Please note, that our study design, hypotheses and analyses were preregistered on OSF (osf.io/5627f).

3. Simple, uncorrected correlations

As noted earlier, an omnibus model (e.g., LMM or even an ANOVA) would allow the authors to directly test for main and interaction effects of stimulus type (vowels vs. phrases) and other relevant factors. Linear models would also allow additional control of effects of speaker ID across conditions. However, the authors have opted to maintain their simple analysis of a series of spearman ranked correlations, without controlling for multiple comparisons. In their response the authors argue that, “we decided to not correct for multiple correlations to avoid false negative conclusions”. Yet without doing so, the authors invite false positives.

At the very minimum, the authors should control for multiple comparisons in **all** of their correlations (rather than a select few, as now presented in supplement, because ‘choosing’ which correlations to correct defeats the purpose). A more sensible solution to the issue of false negatives would be to increase sample sizes, as $N=10$ participants per sex, per experiment, is a very small sample size.

Response #3: We appreciate that false positive results due to multiple comparisons can be a valid and important concern. In the present study, we would have thought that this is less of a concern, to the extent that systematic patterns of findings were obtained in multiple independent data sets. As we had decided to follow the suggestions to collect additional data, we pre-registered a few hypotheses for Exp. 5 that were based on the results in Exp. 1 - 4 (osf.io/5627f). Essentially, we re-assessed the main results in an additional experiment with pre-registered hypotheses and with entirely new speakers and listeners. We hope the consistency and replicability of the main findings convinces the reviewer and editor that our main results and conclusions are clearly not a result of false positives reflecting spurious outcomes of multiple comparisons.

Response to Reviewer's Comments: Manuscript ID: RSOS-190429

Title: "Attractiveness and distinctiveness between speakers' voices in naturalistic speech and their faces are uncorrelated"

Reviewer: 1

Comments to the Author(s)

I thank the authors for their detailed responses to my concerns and for running additional tests, including these as footnotes / exploratory analyses. While I feel that the methods and results retain some issues, that authors have now made reference to these limitations in the paper, thus leaving the judgment to the reader.

Following my earlier comment that has not been fully addressed, I recommend that the authors include stronger rationale (a few sentences) in the Introduction to describe why we might predict average voices to be attractive (from an evolutionary, social, perceptual or any other point of view). While the lack of empirical evidence in the literature linking voice attractiveness and averageness may indeed be tied to methodological obstacles of voice resynthesis, this does not negate the importance of having an a priori prediction as to why such a relationship should exist - and why that is being investigated in the current set of studies.

Response#1: While we did not think that there is a lack of empirical evidence for linking voice attractiveness and averageness (prominent studies we discussed do make that point forcefully, such as Bruckert et al. 2011, Current Biology), we have now considered this suggestion such that we provided a few further sentences to support the link between voice attractiveness and averageness from other perspectives too. In particular, from an evolutionary perspective, average voices may be perceived as more attractive than non-typical voices, in the sense that deviation from the norm could signal ill-health which is thought to be negatively related with perceptions of physical attractiveness (Weeden & Sabini, 2005). For instance, vocal dysphonia (hoarseness, weakness) may be an indicator of unilateral vocal fold paralysis (Eyshold, Rodanowski, & Hoppe, 2003), Parkinson's disease (Little, McSharry, Hunter, Spielman, & Ramig, 2009). From a perceptual point of view, several researchers argued that average (typical) stimuli are processed more fluently and are therefore perceived as more beautiful or likeable than non-typical ones (Reber, Schwarz, & Winkielman, 2004), including voices (Babel & McGuire, 2015). Moreover, a link between averageness and vocal attractiveness is also supported by our own recent data from a larger voice database (Zäske, Skuk, Golle, & Schweinberger, 2020, supplemental materials, Fig. S3). Here we found a moderate, yet significant negative correlation ($\rho = -.342, p < .001$) between attractiveness ratings and acoustic distance-to-mean (i.e., deviation from averageness) in a 3D voice space across 120 young and old male and female speakers.

The new Table S20 (comparison of F0 measured from sentences produced with/without a model) is missing from the supplementary file.

Response#2: This is unfortunate and we are unable to say why. We were sure that we had uploaded the latest version of the supplemental materials, including that table. We will be particularly attentive during the resubmission process to make sure the table is included.

Wishing you all well in these hard times.

Response#3: Thank you, likewise!

Babel, M., & McGuire, G. (2015). Perceptual Fluency and Judgments of Vocal Aesthetics and Stereotypicality. *Cognitive Science*, 39(4), 766-787. doi:10.1111/cogs.12179

- Eyshold, U., Rosanowski, F., Hoppe, U. 2003 Vocal fold vibration irregularities caused by different types of laryngeal asymmetry. *Laryngology*. 260, 412-417. (10.1007/s00405-003-0606-y)
- Little, M. A., McSharry, P. E., Hunter, E. J., Spielman, J., & Ramig, L. O. (2009). Suitability of Dysphonia Measurements for Telemonitoring of Parkinson's Disease. *Ieee Transactions on Biomedical Engineering*, 56(4), 1015-1022. doi:10.1109/tbme.2008.2005954
- Reber, R., Schwarz, N., & Winkielman, P. (2004). Processing fluency and aesthetic pleasure: Is beauty in the perceiver's processing experience? *Personality and Social Psychology Review*, 8(4), 364-382. doi:10.1207/s15327957pspr0804_3
- Weeden, J., & Sabini, J. (2005). Physical attractiveness and health in western societies: A review. *Psychological Bulletin*, 131(5), 635-653. doi:10.1037/0033-2909.131.5.635
- Zäske, R., Skuk, V. G., Golle, J., & Schweinberger, S. R. (2020). The Jena Speaker Set (JESS)-A database of voice stimuli from unfamiliar young and old adult speakers. *Behavior Research Methods*, 52(3), 990-1007. doi:10.3758/s13428-019-01296-0